

# Rain erosivity map for Germany derived from contiguous radar rain data

Franziska K. Fischer[1,2,3], Tanja Winterrath[4], Karl Auerswald[1]

[1]Lehrstuhl für Grünlandlehre, Technische Universität München, Freising, 85354, Germany
5 [2]Bayerische Landesanstalt für Landwirtschaft, Freising, 85354, Germany
[3]Außenstelle Weihenstephan, Deutscher Wetterdienst, Freising, 85354, Germany
[4]Deutscher Wetterdienst, Department of Hydrometeorology, Offenbach/ Main, 63067, Germany

10 *Correspondence to*: Karl Auerswald (auerswald@wzw.tum.de)

**Abstract.** Erosive rainfall varies pronouncedly in time and space. Severe events are often restricted to a few square kilometers. Rain radar data with high spatio-temporal resolution enable this pattern of erosivity to be portrayed for the first time. We used radar data collected with a spatial resolution of 1 km² for 452 503 km² to derive a new erosivity map for Germany and to analyze the seasonal distribution of erosivity. Extraordinarily large filtering was necessary to extract the 15 expected long-term regional pattern from the scattered pattern of events. Filtering included averaging 2001 to 2017 and smoothing in time and space. The pattern of the resulting map generally agreed well with the previous map based on regressions of rain gauge data (mainly from the 1960s to 1980s). The pattern was predominantly shaped by orography. However, the new map has more detail; it deviates in some regions where the regressions previously used were weak; most importantly, it shows that erosivity is about 66% higher than in the map previously used. This increase in erosivity was 20 confirmed by long-term data from rain gauge stations used for the previous map. The change was thus not caused by using a different methodology but by weather changes that may already be a dramatic result of climate change since the 1970s. Furthermore, the seasonal distribution of erosivity showed that more erosivity falls during the winter period when soil cover by plants is usually poor. For many crops higher erosion therefore also results from the change in seasonality. Predicted soil erosion in winter wheat is now about four times higher than in the 1970s due to the seasonal changes, combined with the 25 increased erosivity. These topical erosivity data with high resolution will thus have definite consequences for agricultural advisory services, landscape planning and even political decisions.

## 1 Introduction

Soil erosion by heavy rain is regarded as the largest threat to the soil resource. Rain erosivity, which is a rain's ability to detach soil particles and provide transport by runoff, is one of the factors influencing soil erosion. The most commonly used 30 measure of rain erosivity is the R factor from the Universal Soil Loss Equation USLE (Wischmeier, 1959; Wischmeier and Smith, 1958, 1978), although other concepts also exist (Morgan et al., 1999; Schmidt, 1991; Williams and Berndt, 1977).





The R factor is given as the product of a rain's kinetic energy and its maximum 30-min intensity. Both components are usually derived from hyetographs recorded by rain gauges. Such rain gauge data are spatially scarce. For instance, in Germany only one rain gauge per 2571 km² was available for the R map presently in use (Sauerborn, 1994). Hence, information has to be interpolated to derive an R map that enables R to be estimated for any location. Different interpolation

techniques have been applied. Correlations (transfer functions) to other meteorological data available at higher spatial density were used the most (for an overview see Nearing et al., 2017). The German R-factor map is based on correlations between R and normal-period summer rainfall or normal-period annual rainfall, differing between federal states (Rogler and Schwertmann, 1981; Sauerborn, 1994, and citations therein).

Recent research has shown that the erosivity of single events exhibits enormous gradients in space (Fiener and Auerswald,

2009; Fischer et al., 2016; Fischer et al., 2018; Krajewski et al, 2003; Pedersen et al., 2010; Peleg et al., 2016), which is due to the small spatial extent of convective rain cells typical for erosive rains. The resulting heterogeneity has two consequences. First, interpolation between two neighboring rain stations will not be possible for individual rains because a rain cell in between may be completely missed. Second, even long records of rain gauge data may miss the largest events that occurred in close proximity to a rain gauge and thus underestimate rain erosivity. This is illustrated nicely by the data of

Fischer et al. (2018). They showed that the largest event erosivity, which was recorded by contiguous measurements over only two months, was more than twice as large as the largest erosivity that occurred during 16 years when the same area was covered by 115 rain gauges. Furthermore, this single event contributed about 20 times as much erosivity as the expected long-term average. Even in a 100-yr record this single event would thus still change the long-term average erosivity. The large variability then directly translates to soil loss. This may be illustrated by soil loss measurements in vineyards in

Germany. Emde (1992) found a mean soil loss of 151 t ha$^{-1}$ yr$^{-1}$ averaged over 10 plot years while Richter (1991) only measured 0.2 t ha$^{-1}$ yr$^{-1}$, averaged over 144 plot years. The difference is due to the largest event during the study by Emde (1992), which obviously had too much influence on the mean compared to the size of his data set. Such an event was missing entirely in Richter's (1991) much larger data set. The inclusion of rare events when measured by chance by a rain gauge leads to statistical problems due to their extraordinary magnitude. Unstable and unreliable transfer functions result that differ

pronouncedly depending on whether a large event is included or not. To avoid this, Rogler and Schwertmann (1981) excluded all events for which the estimated return period was more than 30 yr (assuming that event erosivities followed a Gumble distribution), prior to the development of their transfer function. This approach must underestimate erosivity and, in turn, soil erosion because the largest events are then replaced by zero.

The demand for contiguous rain data to create R-factor maps has only recently been able to be met by radar rain data of high

spatial and temporal resolution. Put simply, the measurements are based on the principle that radar beams are reflected by hydrometeors. The intensity of the reflection depends on rain intensity and the travel time of the reflected radar beam depends on the distance between the emitting and receiving radar tower and the hydrometeor. Radars usually measure with a resolution of approx. 1° azimuth and 125 to 250 m in the direction of beam propagation. The data are then typically



transformed to grids of square pixels of 1 km² (Bartels et al., 2004; Fairman et al., 2015), 4 km² (Koistinen & Michelson, 2002; Michelson et al., 2010) or 16 km² (Hardegree et al., 2008) after many refinement steps.

In this study, we used the new RW product from the radar climatology RADKLIM from the German Meteorological Service (Deutscher Wetterdienst, DWD). RW data provide gauge-adjusted and further refined precipitation for a pixel size of 1 x 1 km² (Winterrath et al., 2017, 2018). RW data of 17 yr (2001 – 2017) are available as a contiguous source of rain information. Using these data to establish a new R-factor map for Germany should be a major step forward compared to the existing map, which was derived from an inconsistent set of data compiled by different researchers (e.g., some had winter precipitation data available and used it while others did not; see Sauerborn, 1994) and with equations developed independently for 16 federal states. Our data set is much larger (by a factor of 2571 regarding locations) and, because of the contiguous data source, it does not require interpolation with transfer functions. We expect that there will be considerable changes in the pattern of erosivity due to the removal of transfer-function weaknesses. We also expect that the R-factor map will exhibit higher values than the existing map, for two reasons. Very large and rare events will no longer be missed, as occurred previously due to the large distances between meteorological stations, and there is no longer any need to remove these events to arrive at robust transfer functions. The second reason for higher R factors is due to global climate change, as Rogler and Schwertmann (1981) and Sauerborn (1994) mostly used data from the 1960s, 1970s and 1980s. Global climate change is expected to increase rain erosivity (Burt et al., 2016).

## 2 Material and methods

### 2.1 Radar-based precipitation data

DWD runs a Germany-wide network of, at present, 17 C-Band Doppler radar systems (Fig. 1). This network underwent several upgrades during the analysis period. At the start of the time period considered, five single-polarization systems (DWSR-88C, AeroBase Group Inc., Manassas, USA) were operated without a Doppler filter, the latter being added between 2001 and 2004. Between 2009 and 2017, DWD replaced the network of C-band single-polarization systems of the types METEOR 360 AC (Gematronik, Neuss, Germany)  and DWSR-2501 (Enterprise Electronics Corporation, Enterprise, USA) with modern dual-polarization C-band systems of the type DWSR-5001C/SDP-CE (Enterprise Electronics Corporation), all equipped with Doppler filters. During this period, a portable interim radar system of the type DWSR-5001C was installed at some sites.

The radar systems permanently scan the atmosphere to detect precipitation signals. Every five minutes, the radars perform a precipitation scan, each with terrain-following elevation angle to measure precipitation near the ground. The resulting local reflectivity information over a range of 128 km is combined to form a Germany-wide mosaic of about 1100 km in the north-south and 900 km in the west-east direction. The reflectivity information is converted to precipitation rates applying a reflectivity–rain rate (ZR) relationship. An operational quality control system screens the radar data. To further improve the quantitative precipitation estimates, the radar-derived precipitation rates are summed to hourly totals and immediately



adjusted to gauge data resulting in RADOLAN (i.e. online-adjusted, radar-derived precipitation), which provides precipitation data in real time, mainly for applications in flood forecasting and flood protection (Bartels et al., 2004; Winterrath et al., 2012).

Based on RADOLAN, the climate version RADKLIM is derived. Compared to the real-time approach, the data are
additionally offline-adjusted to daily gauge data, combining a total of more than 4400 rain gauges measuring hourly and daily (1 rain gauge per 80 km²). The data are then reprocessed by new climatological correction methods, e.g. for spokes, clutter or short data gaps. Spokes result from permanent obstacles blocking the radar beam, while clutter is introduced by non-meteorological targets like windmills or birds. The final product (RW data) has a temporal resolution of 1 h and a spatial resolution of 1 km x 1 km in polarstereographic projection. For more detailed information on RADKLIM the reader is
referred to Winterrath et al. (2017). The RW data, restricted to the German territory, are freely available (Winterrath et al., 2018). For the first time, the RADKLIM data set provides contiguous precipitation data with high temporal and spatial resolution. It includes local heavy precipitation events that are partly missed by point measurements alone. Thus, it particularly improves the analysis of extreme precipitation events.

Two additional data sets were used to verify the validity of the approach and to examine effects of methodological details
(see below). These data sets are erosivities derived from radar data at 5-min resolution taken from Fischer et al. (2016) and erosivities derived from 115 rain-gauge station data in Germany during 2001 to 2016, which were taken from Fischer et al. (2018).

## 2.2 Erosivity calculation procedures

According to Wischmeier (Wischmeier, 1959; Wischmeier and Smith, 1958, 1978), the erosivity of a single rain event ($R_e$) is
the product of the maximum 30-min rain intensity ($I_{max30}$) and the total kinetic energy ($E_{kin}$). For hyetographs recorded by rain gauges, an erosive rain event is defined as a total precipitation amount ($P$) of at least 12.7 mm or an $I_{max30}$ of more than 12.7 mm h⁻¹ that is separated from the next rain by at least six hours.

$$R_e = I_{max30} * E_{kin} \tag{1}$$

Kinetic energy $E_{kin,i}$ per mm rain depth (in kJ m⁻² mm⁻¹) is given for intervals $i$ of constant rain intensity $I$:

$$E_{kin,i} = (11.89 + 8.73 * log_{10}I) * 10^{-3} \qquad \text{for } 0.05 \text{ mm h}^{-1} \leq I < 76.2 \text{ mm h}^{-1} \tag{2.1}$$

$$E_{kin,i} = 0 \qquad \text{for } I < 0.05 \text{ mm h}^{-1} \tag{2.2}$$

$$E_{kin,i} = 28.33 * 10^{-3} \qquad \text{for } I \geq 76.2 \text{ mm h}^{-1} \tag{2.3}$$

For all intervals $i$, $E_{kin,i}$ is multiplied with the rain amount of this interval and then summed to yield $E_{kin}$ for the entire event. The annual erosivity of a specific year is the $R_e$ sum of all $n$ erosive events within this year. The average annual erosivity ($R$)
is then the average of all annual erosivities during the study period (17 yr in this case). While in the USA and other countries the unit MJ mm ha⁻¹ h⁻¹ is often used for $R_e$, we use N h⁻¹ because it is the unit most often used in Europe and because of its




simplicity. Both units can be easily converted by multiplying the values in N h$^{-1}$ with a factor of 10 to yield MJ mm ha$^{-1}$ h$^{-1}$. The unit for $R$ is then N h$^{-1}$ yr$^{-1}$.

Rain erosivity strongly depends on intensity peaks. Fischer et al. (2018) have shown that these peaks increasingly disappear the lower the spatial and temporal resolution becomes. This can be accounted for by scaling factors but these scaling factors

can only adjust to an average behavior, while the influence of the true event $R_e$ may either be too large or too small. A high spatio-temporal resolution should be used to determine $R_e$ for individual events. To determine the long-term average pattern, i.e. an R-factor map for planning and prediction purposes, using data with lower resolution and applying appropriate scaling factors is advantageous because this will reduce the noise introduced by large events of small spatial extent that would not be leveled out by averaging alone. We will use data in 1-h time increments, which additionally have the advantages that they

are adjusted to rain gauge measurements and the amount of data is reduced by a factor of 12 compared to 5-min increments. This is especially important when all calculations, including identification of rain breaks > 6 h and periods of $I_{max30}$, have to be carried out for many years and many locations. In our case, roughly $7 \times 10^{10}$ 1-h increments had to be processed.

Gaps in the time series have been considered when calculating mean R factors by scaling the total sum of erosivity over the whole time series to 365.25 days. If the effective number of missing values exceeded two months per year, the respective

15  year was excluded from the calculation for that pixel.

According to Fischer et al. (2018), the following modifications in the calculation of $R_e$ had to be made to account for the temporal resolution of 1 h, the spatial resolution of 1 km² and radar measurement: (i) $I_{max30}$ was replaced by the maximum 1-h rain depth and the threshold was lowered to 5.8 mm h$^{-1}$, while the total precipitation threshold remained at 12.7 mm. (ii) Rain breaks separated events when five subsequent 1-h intervals without rain occurred. This assumed that rain events stop

and start on average in the middle of the first and the last non-zero rain interval, yielding a total rain break of 6 h. (iii) The temporal scaling factor was 1.9 and the spatial scaling factor was 1.13, to which 0.35 has to be added to account for the radar measurement instead of the rain gauge measurement. The total scaling factor $[(1.13 + 0.35) \times 1.9]$ was then 2.81.

## 2.2 Generating a Germany-wide R-factor map

The reduction of noise by using 1-h increments was still not sufficient to level out the most extreme events. Two further

filtering steps were therefore applied, in addition to using a 17-yr mean. The first averaging step was to winsorize the annual erosivities of the 17 yr (Dixon and Yuen, 1974) for each individual pixel by replacing the lowest value with the second-lowest value and the highest value with the second-highest value. Winsorizing is an appropriate measure for calculating a robust estimator of the mean in symmetrically distributed data but it is biased for long-tailed variables like rain erosivity. Thus, the country-wide mean of all winsorized data (96 N h$^{-1}$ yr$^{-1}$) was lower than the mean of the original data (98 N h$^{-1}$

30  yr$^{-1}$). In order to remove this bias, we binned all data in 26 bins of 20 N h$^{-1}$ yr$^{-1}$ width and calculated the mean R before and after winsorizing. For bins with R < 180 N h$^{-1}$ yr$^{-1}$, comprising 95% of all pixels, the bias increased linearly with R (r² = 0.92; n = 8) and amounted to 2.3% of R. Above 180 N h$^{-1}$ yr$^{-1}$ there was no further increase in the bias (r² = 0.01, n = 18),



which was, on average, 3.4 N h$^{-1}$ yr$^{-1}$. We removed the bias by adding 2.3% to all values < 180 N h$^{-1}$ yr$^{-1}$ and 3.4 N h$^{-1}$ yr$^{-1}$ to all values above.

The third noise-reduction step applied geostatistical methods. A semivariogram (over a range of 50 km) was calculated and ordinary kriging was applied. Geostatistical analysis was done in R (version 3.5.0; R Core Team, 2018) using gstat (Gräler et al., 2016). To remove noise, a block size of 10 × 10 km² was chosen, while the spatial resolution remained at 1 km. This step was also necessary to fill pixels with data gaps of more than one year (0.6% of the entire area). The missing information was obtained from neighbor pixels. The radar data extended beyond German borders. In total, 452 503 pixels were used to ensure low krige errors near borders or on islands, while the final map was restricted to the German land surface (357 779 pixels).

Using 1-h data instead of 5-min data reduced the effect of single extreme events at certain locations. Winsorizing reduced the effect of extreme years at a location, in addition to the effect of averaging 17 yr. Finally, kriging used the information from neighbor pixels to reduce the effect of the extremes. This should not have affected the regional pattern. To evaluate whether this was the case and to quantify the effect of all smoothing steps, we used the data from Fischer et al. (2016), who calculated rain erosivity from 5-min-resolution radar data for two years (2011 and 2012) and an area of 14 358 km² (yielding a total of 28 770 pixel years), called "test region" in the following. Using these data we calculated semivariograms after each smoothing step from annual to biennial erosivities based on 5-min and 1-h resolution, for 17-yr average erosivities, for winsorized averages and finally for kriged values. Smoothing should reduce the influence of individual violent thunderstorm cells and reveal the regional pattern. In geostatistical analysis this decreases the sill of the semivariogram while the range increases as it changes from being dominated by thunderstorm cells to being dominated by the regional pattern.

## 2.3 Return periods

Rain erosivity usually follows long-tailed distributions, which leads to the question of how frequent years of extraordinarily large erosivity are, which requires the development of cumulative distribution curves (for basic concepts see Stedinger et al., 1993). Seventeen years are not sufficient to reliably estimate a cumulative distribution curve for every pixel. We combined all data (452 503 pixels and 17 yr) after expressing the event erosivities of all individual years relative to the winsorized and bias-corrected mean of each pixel (in percent). This enabled the cumulative distribution curves to be calculated from a large data set (n = 7.7 million) and the expected maximum relative annual erosivity for a given return period to be estimated from the complementary cumulative distribution curve (exceedance). This was also done for the relative annual erosivities of the test region, calculated from 1-h rain data, to examine whether the general cumulative distribution curve also applies to smaller regions.

The erosivities, when calculated from 1-h rain data, are already smoothed and do not adequately reflect the extremes that result from data that are more highly resolved, such as the 5-min rain data. The cumulative distribution curve for the test region was also calculated using the 5-min rain data. Given that the cumulative distribution curves of the entire study area and the test region agree for the relative erosivities calculated from 1-h data, this can also be expected to be the case for the



relative erosivities calculated from 5-min rain data. The cumulative distribution curve for the test region calculated from 5-min data will then be a fair estimate of the return periods anywhere in the research area.

## 2.4 Annual cycle of rain erosivity

The seasonal variation, calculated as the relative contribution of each day to total erosivity, is called erosion index distribution or EI distribution (Wischmeier and Smith, 1978). It is required in erosion modeling to determine the influence of seasonally varying soil cover due to crop development. The convolution of the seasonal effect of soil cover with the seasonal EI distribution results in the so-called crop and cover factor (C factor). The EI distribution was calculated for each pixel and averaged over all 452 503 pixels. Seventeen years of data still did not suffice to show similar amounts of erosivity on subsequent days, despite the large number of pixels. There was still considerable scatter that required smoothing to illustrate the seasonal distribution. Smoothing between individual days during the year involved three steps (for details of the methods see Tukey, 1977): first a 13-d centered median was calculated for each day. A centered median preserves the common trend signal and the level shifts in the smooth (Gallagher and Wise, 1981), which is also true for the two subsequent steps. A 3-d skip mean (leaving out the second day) was calculated from the results, followed by a 25-d centered hanning mean (weighted mean with linearly decreasing weights). The year was recycled to allow the smoothing methods to be applied at the boundaries.

The EI distribution deviated from the EI distribution used previously. This was especially pronounced during the winter months. However, radar measurements tend to have larger errors during wintertime with snowfall. The reduced reflectivity of snow particles may lead to an underestimation of the precipitation rate, while the increased reflectivity of melting particles in the bright band may cause on overestimation. Moreover, the lower boundary layer promotes a potential overshooting of the radar beam with regard to the precipitating cloud. Therefore, we also calculated the EI distribution using data from 115 rain gauges distributed throughout Germany and covering 2001 to 2016. These data were taken from Fischer et al. (2018). This data set will also be used in the discussion for comparison of recent radar-derived erosivities with recent raingauge-derived erosivities and with historic raingauge-derived erosivities taken from literature.

## 3 Results

### 3.1 Erosivity map

The regional pattern (Fig. 2) was mainly determined by orography. Highest values (above 185 N h$^{-1}$ yr$^{-1}$) were found in the very south where the northern chain of the Alps reaches altitudes of almost 3000 m above sea level (a.s.l.). Smaller mountain ranges are also characterized by high mean annual erosivities. For instance, the Bavarian Forest, in the southeast on the Czech border with elevations of up to 1450 m a.s.l., exhibited annual erosivities of above 155 N h$^{-1}$ yr$^{-1}$. The Ore Mountains in the east also on the Czech border, with elevations of up to 1244 m a.s.l., had erosivities mostly between 125 and 155 N h$^{-1}$ yr$^{-1}$. Also mountain ranges like the Black Forest (a mountain range in southwestern Germany oriented north-south) or the



Harz Mountains (an area of high erosivities located almost in the middle of northern Germany) clearly shape the erosivity map. Upwind-downwind effects were detectable. For example, the areas west-north-west (upwind) of the Harz Mountains had erosivities of between 70 and 80 N h$^{-1}$ yr$^{-1}$, while the areas east-south-east (downwind) received less than 65 N h$^{-1}$ yr$^{-1}$.

### 3.2 The effects of smoothing

Winsorizing reduced the standard deviation (SD) of a pixel over time from, on average, 49 N h$^{-1}$ yr$^{-1}$ to 39 N h$^{-1}$ yr$^{-1}$, while bias correction left the mean for all pixels unchanged at 98 N h$^{-1}$ yr$^{-1}$. Using kriging, the mean remained the same at 98 N h$^{-1}$ yr$^{-1}$, because kriging is an unbiased linear interpolator that smoothed locally (over distances of about 10 to 15 km). Only the very extreme values disappeared. Values lower than 45 N h$^{-1}$ yr$^{-1}$, which had contributed 0.06% to the winsorized data, disappeared. In addition, values larger than 450 N h$^{-1}$ yr$^{-1}$, which had contributed 0.03% to the winsorized data, also
disappeared.

Rain erosivity from 5-min resolution data for the test region showed large small-scale variability, even for annual sums of erosivity (Fig. 3, upper panel). The range was only 20 km, indicating that the annual pattern was dominated by individual cells of violent rain. The semivariance for a lag of 20 km was 2749 N$^2$ h$^{-2}$ yr$^{-2}$ (Table 1). Using the normal distribution, in 31.8% of all cases the difference between two pixels separated by 20 km must then be larger than 52 N h$^{-1}$ yr$^{-1}$ (square root
of 2749 N$^2$ h$^{-2}$ yr$^{-2}$), which is more than half the average annual erosivity in Germany. After averaging both years (2011 and 2012), the semivariance for a lag of 20 km was reduced to 1569 N$^2$ h$^{-2}$ yr$^{-2}$ and the range stayed the same at approximately 20 km. Both findings indicated that even after averaging two years, the individual cells of violent rain were still fully detectable and had not merged to form a larger pattern.

The effect when using data with a resolution of 1 h was almost as strong as when two years were averaged. Semivariance at
a lag of 20 km was only 1667 N$^2$ h$^{-2}$ yr$^{-2}$ for annual values and 953 N$^2$ h$^{-2}$ yr$^{-2}$ for biannual averages. Even more importantly, due to smoothing of the extreme events, the regional trend became better visible. This trend is evident from the gradual increase in semivariance over the entire range of lags shown in Fig. 3. This regional trend was already detectable in the annual erosivities calculated from 5-min data (Table 1), but did not appear to be significant due to the large semivariance caused by cells of violent rain. Importantly, smoothing using 1-h data did not change overall erosivity. The biannual average
for the test region was 115 N h$^{-1}$ yr$^{-1}$ when calculated from 5-min data and 114 N h$^{-1}$ yr$^{-1}$ when calculated from 1-h data.

Averaging 17 yr further reduced variability (Fig 3, upper panel). Semivariance strongly decreased to 197 N$^2$ h$^{-2}$ yr$^{-2}$ and the influence of individual cells of violent rain became small relative to the regional trend, leading to an almost linear increase in semivariance over distance. The influence of extraordinary years in individual pixels was further reduced by winsorizing, which slightly reduced semivariance at 20 km distance to 190 N$^2$ h$^{-2}$ yr$^{-2}$.

Finally, kriging reduced semivariance at 20 km distance to 121 N$^2$ h$^{-2}$ yr$^{-2}$, leaving mainly the regional trend. Thus, the step from 5-min to 1-h resolution reduced semivariance at 20 km by a factor of 1.6; averaging 17 yr reduced semivariance by a factor of 8.5; winsorizing contributed a factor of 1.04 and kriging a factor of 1.6. In total, semivariance was reduced by a factor of 23, indicating a pronounced patchiness of erosive rains on the annual scale that could not be leveled out by



averaging 17 years alone. These factors became larger at shorter distances (e.g. the combined factor was 32 for a lag of 10 km) because the importance of thunderstorm cells, relative to the regional trend, increased. Correspondingly, the combined effect decreased with increasing distance (e.g., the factor was only 13 for a lag of 40 km) because the regional trend, which was not removed by the smoothing procedures, became increasingly important. The regional trend, extracted from the change in semivariance between lags of 20 km and 40 km, remained practically unchanged at 0.2 N h$^{-1}$ yr$^{-1}$ km$^{-1}$, independent of the degree of smoothing (Table 1). In contrast, the effect of violent rain cells decreased greatly from 2.4 N h$^{-1}$ yr$^{-1}$ km$^{-1}$ to 0.3 N h$^{-1}$ yr$^{-1}$ km$^{-1}$.

After winsorizing and kriging, the semivariances for the test region followed a linear regression through the origin almost perfectly (r² = 0.9889, n = 50). This indicated that the variation in erosivity over a distance of 50 km followed linear trends without any noise (nugget) or small-range structures that could be attributed to individual cells of violent rain. The semivariances, when calculated for the whole of Germany, were considerably higher (twice as high at a lag of 50 km; Fig. 3, lower panel) and close to a linear trend only for short distances (e.g. a linear regression through the origin for the first 15 km yielded r² = 0.9905). For larger distances, the semivariogram followed an exponential model (nugget 4 N$^2$ h$^{-2}$ yr$^{-2}$, partial sill 970 N$^2$ h$^{-2}$ yr$^{-2}$, effective range 123 km). The inclusion of mountain areas with high erosivities and steep erosivity gradients that were missing in the test region led to both the higher semivariance and the exponential model.

### 3.3 Return periods

The cumulative distribution of the relative annual erosivities followed a straight line in a probability plot fairly well when the logarithm was used (Fig. 4), indicating a log-normal distribution (log mean 1.96; log SD 0.19). A very similar cumulative distribution was found for annual values derived from the 1-h data of the test region (log mean 1.97; log SD 0.18), while the distribution was considerably wider for the less-smoothed 5-min data (log mean 1.94; log SD 0.22). The annual expected erosivity was 88%, 216% and 273% of the respective mean for return periods of 2 yr, 30 yr, and 100 yr when the 5-min data were used (Fig. 4). It is important to note that these values apply for averages of 1 km² pixels and include the smoothing that results from the radar measurement, the radar reprocessing, and from using 5-min rain increments. Even extremer years are expected to occur in reality.

### 3.4 Erosion index distribution

There was a pronounced peak in the relative seasonal variation during summer months (Fig. 5). The relative daily erosion index increased rapidly from mid-April to mid-May to a mean of 0.61 % d$^{-1}$ in June, July and August, and declined rapidly again from mid-August to September. The contribution of winter months was small (mean of December, January, February, and March: 0.08 % d$^{-1}$). Even more striking was the fact that this pattern required considerable smoothing to yield a continuous seasonal time course. The difference between subsequent days in the unsmoothed data was enormous (e.g., 1.5 % d$^{-1}$, 0.4% d$^{-1}$ and 0.4% d$^{-1}$ on July 29, 30 and 31). This was despite the large number of measurements (17 yr and 455 309 pixels) that were averaged for each day. It highlights how extreme some violent rains must be. Despite the rather small



extent of individual erosivity cells, many of them occurred at the same day making a high relative contribution to total erosivity for this day. While particular days of the year are influenced by heavy precipitation, during other days no erosive rain fell anywhere within the research area. Seventeen years were not sufficient to level out the contrast between subsequent days. The results of the smoothing procedure show that even 221 yr (17 yr multiplied by a moving-average window of 13 d) would not be sufficient to level out these differences because two additional smoothing steps had to be applied to arrive at a smooth time course.

The EI distribution, when calculated from rain gauge data (1840 station years), was very similar to the distribution calculated from the much larger radar data set. This was especially true during winter months, when values derived from both measurement methods were considerably higher than expected.

## 4 Discussion

### 4.1 Increase in erosivity

The most striking difference between the German R map presently in use (Sauerborn, 1994) and the radar-derived map is a pronounced increase in erosivity. A German average of 58 N h$^{-1}$ yr$^{-1}$ was derived from the Sauerborn map (1994) (Auerswald et al., 2009), while the radar-based map suggests an average of 98 N h$^{-1}$ yr$^{-1}$. This will increase predicted soil losses by 69%. An almost identical increase resulted when the erosivity of meteorological stations, as reported by Sauerborn (1994), was compared with the erosivity derived from radar data at the same locations, which resulted in an increase of 63% (Fig. 6). Thus, the increase in erosivity is not an effect of the regression approach that was previously used or due to better capturing of extreme events by the contiguous radar data.

Fischer et al. (2018) calculated erosivity for 33 of the Sauerborn stations from recent (2001 to 2016) rain gauge data. A comparison of these data with the Sauerborn data (1994) also showed a similar increase of 52% (Fig. 5). The increase in erosivity between the Sauerborn map (1994) and the new radar-derived map is thus also not an artifact of using radar data but the result of a true change in erosivity over time. This is further corroborated by Fiener et al. (2013), who analyzed long-term records from ten meteorological stations in western Germany. They found an increase in erosivity of 63% between 1973 and 2007. Both independent findings leave little doubt that the pronouncedly higher values in the new erosivity map are a result of a change in weather properties and not a result of the difference in the applied methodologies, although we did expect the mean to increase due to the contiguous data set, which is better at recording rare extremes.

### 4.2 Change in the regional pattern

The regional patterns of the Sauerborn erosivity map (1994) that is currently used, and that of the radar-based map, generally agree well, but with two exceptions. First, the radar-based map shows distinctly higher values south-east of the German Bight where the air masses coming from the North Sea are channeled by the Elbe river estuary and its Pleistocene meltwater valley and then hit the higher areas of the North German moraines. A high frequency of large rains is not unlikely in this





situation. The reason that this was missed by Sauerborn (1994) using the data obtained by Hirche (1990) for Lower Saxony is mainly due to the low data density and the regression approach. Only 18 stations were available for the whole of Lower Saxony and only five of them were in the area of high erosivity. Using the 18 stations in the state of Lower Saxony only, and ignoring the difference between landscapes, resulted in a rather poor regression with long-term annual rainfall ($r^2$ was only

0.32), and therefore a large prediction error and considerable smoothing of the true erosivity pattern can be expected. For comparison: in Bavaria the regression with long-term rainfall yielded $r^2$ of 0.92 (Rogler and Schwertmann, 1981).

The second difference in the pattern is that the radar-derived map reveals more detail than the regression-based map by Sauerborn (1994). This is especially evident in southern Germany where southwest-northeast oriented structures seem to follow thunderstorm trains. In the north-east quarter of Germany, where the pattern is not shaped by mountain ranges, a

rather patchy pattern resulted. Although Sauerborn (1994) had already found a patchy pattern in this area it appears to be patchier now. It is, at present, difficult to decide whether this pattern is random due to large multi-cell clusters of rainstorms and will level out on the long term, or whether landscape properties, e.g. the existence of large forests, cause a stable pattern in an area where other factors affecting the pattern are missing. More detailed variation may also be expected in mountainous areas but radar measurements cannot adequately show this variation. In the future, using data obtained by commercial

microwave links as an additional source for retrieving precipitation (Chwala et al., 2012, 2016; Overeem et al., 2013) may improve high resolution estimates, particularly in these areas.

## 4.3 Change in the seasonal distribution of the erosion index

The third pronounced difference was found for the erosion index distribution needed for C-factor calculations. A change in the seasonality of erosivity was already suggested by Fiener et al. (2013) analyzing an 80-yr time series. However, Fiener et

al. (2013) used data from April to October only, and their results therefore cannot be compared directly with our results that show the most pronounced changes for the period from December to March.

At present, only the erosion index developed by Rogler and Schwertmann (1981) for Bavaria is used for C-factor calculations in Germany (e.g. Schwertmann et al., 1990; DIN, 2017), although unpublished erosion indices are also available for other federal states (e.g., Hirche, 1990). The index by Rogler and Schwertmann (1981) is characterized by very low

values during winter months, which in turn causes a sharp increase during summer months. In contrast, the radar-based index, although still having a pronounced summer maximum, predicts a higher percentage of erosivity during winter. Rogler and Schwertmann (1981) found that only 1.5% of the annual erosivity fell from January to March, while Fig. 5 indicates that these months contributed 6.9% to annual erosivity. This deviation may be caused by a regional variation in the erosion index because the unpublished indices for other federal states also suggested a higher contribution by winter months (e.g., January

to March contributed 7.5% in Lower Saxony according to Hirche, 1990). However, restricting our data set to Bavaria led to a very similar index during winter months (e.g., 6.2% for January to March) to the index for the whole of Germany and the discrepancy with Rogler and Schwertmann (1981) remained.



A second explanation is that the Rogler and Schwertmann data (1981) were too limited to capture enough erosive rains. This explanation is corroborated by the large scatter between individual days that still exists in our data set (Fig. 5), although our data set was more than 50 000 times larger than the data set used by Rogler and Schwertmann (1981).

A third explanation could again be climate change. In Germany extreme rainfall events have increased in winter by 463% during the last century with the trend greatest during the last few decades, while summer and autumn remained unchanged (Schönwiese et al., 2003).

The change in erosion index may be regarded as being rather unimportant at first glance because erosivity is still dominated by precipitation in summer. This small increase in erosivity during the winter months, however, could have important consequences for the C factor of crops that, due to their crop development stage, provide only a small amount of soil cover during the winter. As there is practically no growth during winter, these crops are susceptible to erosion over a long period and thus experience a considerable amount of erosivity, even though erosivity per day is low. Calculating the C factor for continuous winter wheat from the soil loss ratios and cropping-stage dates reported by Schwertmann et al. (1990) yields a C factor of 0.04 if the erosion index from Rogler and Schwertmann (1980) is used. The C factor increases to 0.10 when the erosion index in Fig. 5 is applied. Thus, the predicted soil loss for continuous wheat is more than twice as high as previously expected due to the change in the erosion index (and four times higher if the change in erosivity is also considered). While the C factor of the maize year in a typical maize-winter wheat rotation is currently regarded to be eight times higher than that of winter wheat, it is only four times higher when the new erosion index is applied. Furthermore, the C factor of the entire rotation increases by 15%.

## 4.4 Stochasticity

Soil erosion is characterized by a large temporal variability at a small spatial scale due to the stochastic character of erosive rains. About 20 yr are necessary, according to Wischmeier and Smith (1978), until this variability levels out and average soil loss approaches values predicted with the (R)USLE. Our data set covered 17 yr but significant additional smoothing was still necessary. This implies that 20 yr will still not be sufficient to level out extraordinary events. Most studies measuring soil erosion under natural rain use much shorter intervals that usually cover only a few years and rarely exceed ten years (see Auerswald et al., 2009, for a meta-analysis of German studies and Cerdan et al., 2010, for European studies). The interpretation of such short-term studies and the applicability of the results are limited due to the pronounced variability of natural rains.

We applied stepwise smoothing in order to minimize the disadvantages inherent in different smoothing procedures and to be able to smooth in time and space. We used rain data with hourly resolution instead of more highly resolved data and compensated for the disadvantages by using a scaling factor and by adjusting the threshold intensity according to Fischer et al. (2018). This procedure smoothed between individual rains. We added winsorizing and bias correction to smooth between years at a certain location. Finally, we added geostatistical smoothing to level out differences between neighboring locations caused by the small spatial extent of erosive rain cells. While the effects of winsorizing and geostatistical smoothing are



rather easy to assess, the effects are less clear when hourly resolved data are used, although the mean erosivity was identical for our test region when calculated with both resolutions. The pronounced influence of orography on the R-factor map could have, at least partly, been caused by this smoothing step. Orographic rainfall may increase hourly rainfall but it may not, to the same degree, increase high intensity peaks that exert a dominant influence on erosivity. Still, this is presently speculative

because the high variability of erosive events means that this question cannot be answered using a 17-yr time series. In one or two decades the data series may be long enough to use data of 30-min or even 5-min resolution.

In addition, the erosion index required pronounced smoothing to improve representation of the seasonal variation. The shift of a certain crop stage by only one day can cause large discrepancies in the resulting C factor, depending on whether a day of large erosivity in the past is included or excluded at the bounds of the crop stage. Smoothing can prevent this. This is

especially important for short crop stages, while the effect becomes small for longer periods. For instance, the monthly sums of the smoothed data correlated closely with the sums of the unsmoothed data (coefficient of determination: 0.995; Nash-Sutcliffe efficiency: 0.994).

## 5. Conclusions

Radar-derived rainfall enables highly resolved and contiguous maps of erosivity to be derived. This yielded a rain erosivity

map with high spatial detail and avoided errors in landscapes with insufficient rain gauge density. The data showed that present (2001 to 2017) rain erosivity is considerably higher than previously expected. Furthermore, the seasonal distribution of rain erosivity also deviated from current expectations and indicated that winter months make a higher contribution to total erosivity than previously thought. Considerably more erosion can be expected for crops that are at a highly susceptible stage of development in winter. In consequence, the predicted soil loss will change pronouncedly by using radar-derived erosivity

and the ranking of crops regarding their erosion potential will change. This will have definite consequences for agricultural extension and advisory services, landscape planning and even political decisions.

Author contribution

KA designed the analysis, which was mainly carried out by FF. TW provided most data and the knowledge about all steps

involved in radar data creation. KA prepared the manuscript with contributions by FF and TW.

## Acknowledgements

This study was part of the project "Ermittlung des Raum- und Jahreszeitmusters der Regenerosivität in Bayern aus radargestützten Niederschlagsdaten zur Verbesserung der Erosionsprognose mit der Allgemeinen Bodenabtragsgleichung" at the Bavarian State Research Center for Agriculture (PI Robert Brandhuber). It was funded by the Bayerisches

Staatsministerium für Ernährung, Landwirtschaft und Forsten (A/15/17). The RADKLIM data were from the project



"Erstellung einer dekadischen radargestützten hochauflösenden Niederschlagsklimatologie für Deutschland zur Auswertung der rezenten Änderung des Extremverhaltens von Niederschlag (Kurztitel: Radarklimatologie)" financed by the Strategic Agencies' Alliance "Adaptation to Climate Change" consisting of the Federal Office of Civil Protection and Disaster Assistance (BBK), the Federal Institute for Research on Building, Urban Affairs and Spatial Development (BBSR), the

Bundesanstalt Technisches Hilfswerk (THW), the Umweltbundesamt (UBA), and the Deutscher Wetterdienst (DWD). Melanie Treisch helped with ArcGIS operations, Karin Levin provided language editing, and Helmut Rogler has, for many years, requested that the German R-factor map be updated.

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



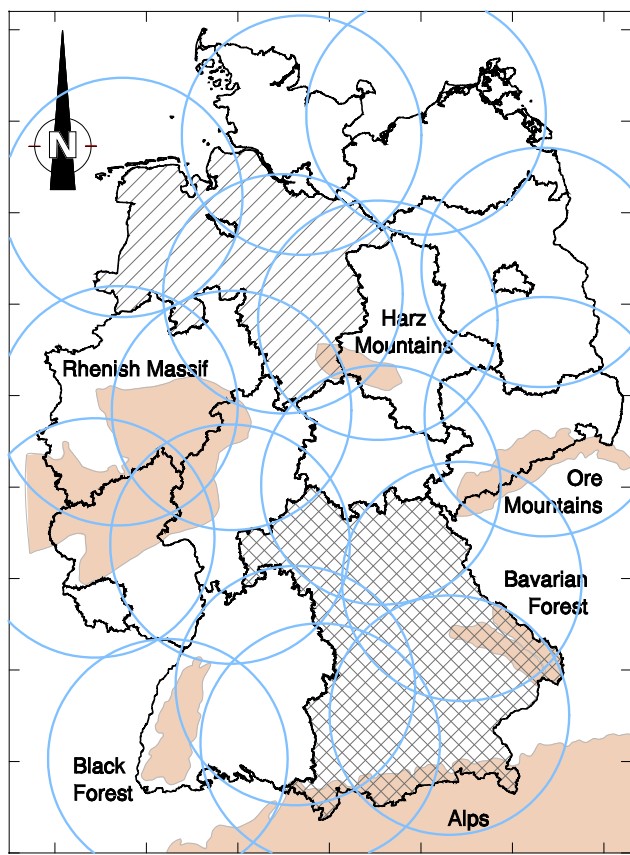

**Figure 1: Coverage (blue circles) of the 17 German rain radars for a range of 128 km and the 2017 configuration (locations of some radar towers have changed over time). Black lines denote federal states; the federal states of Bavaria (cross-hatched), Lower Saxony (hatched) and selected mountain ranges (light brown) are mentioned in the text. Axis ticks represent distances of 100 km.**





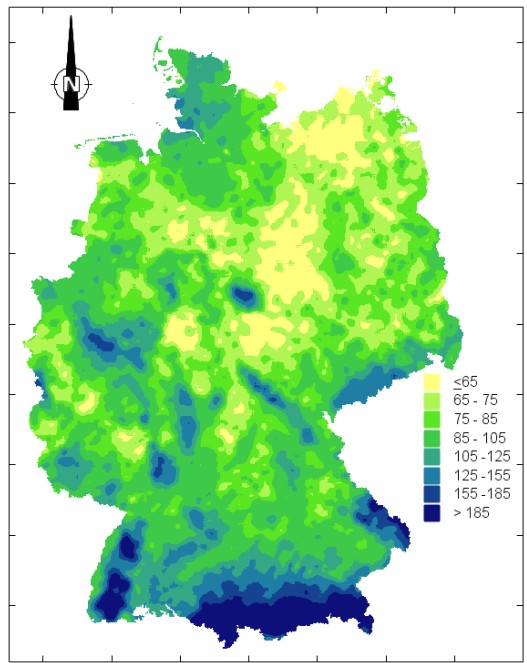

**Figure 2: Erosivity map of Germany from 17 yr of radar rain data. Axes ticks represent distances of 100 km. Color classes from yellow to dark blue comprise approximately 10%, 20%, 20%, 25%, 15%, 4%, 3%, and 3% of the area, respectively. For comparison with the map before winsorizing and before kriging see Figs. A1 and A2 in the Appendix. For average values for the 294 local authority areas (average size 1214 km²), see Table A1 in the Appendix. Average values for the 11 254 communities (average size 32 km²) can be obtained at https://www.lfl.bayern.de/iab/index.php.**



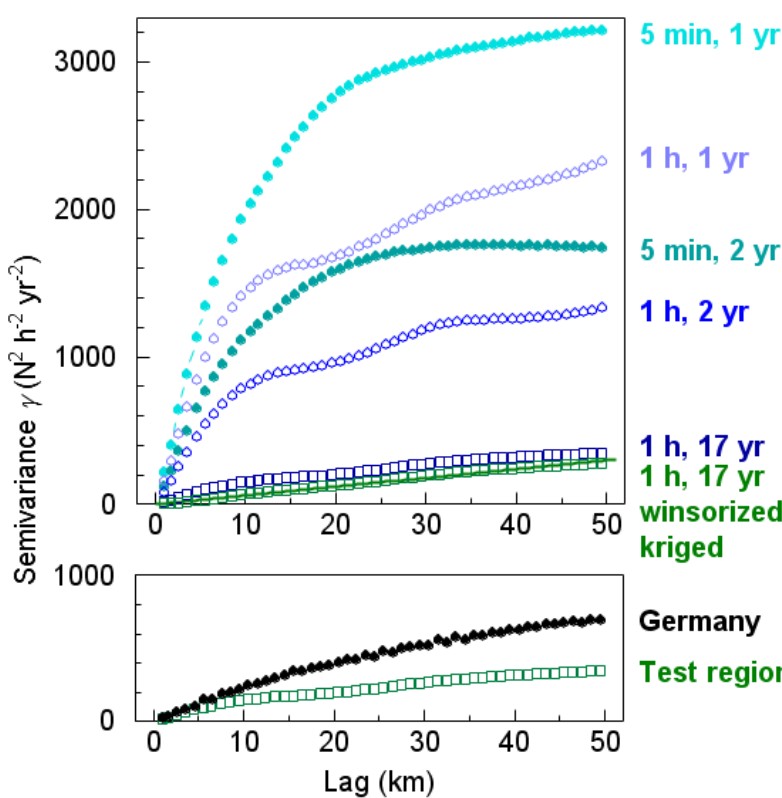

**Figure 3: Upper panel: Semivariograms of erosivity for different temporal resolutions of rain data (5 min and 1 h), different averaging (1 yr, 2 yr, 17 yr), winsorizing and kriging for the test region (for selected lag classes see Table 1). The line through the semivariances of the 1 h, 17 yr, winsorized and kriged data is a linear regression through the origin (r² = 0.9889).**

5    **Lower panel: Comparison of semivariances for the 1 h, 17 yr and winsorized data before kriging for the test region and for the whole of Germany.**





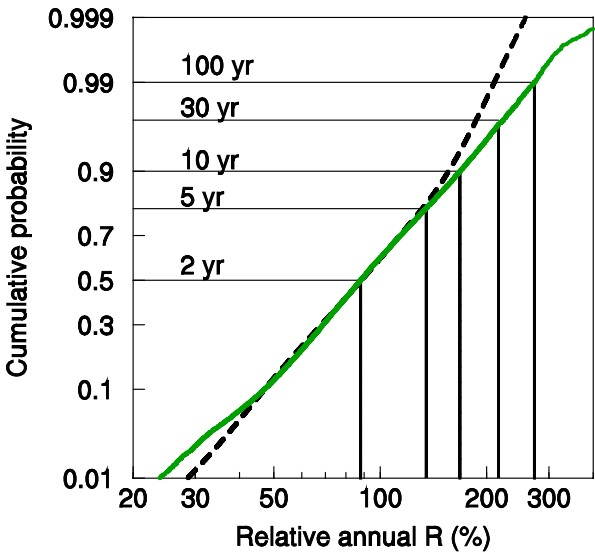

**Figure 4: Cumulative distribution curve of the annual R factor relative to the long-term annual mean of a pixel. Dashed black line applies for erosivities derived from 1-h data for the whole of Germany and 17 yr (n = 7.7 million). Solid green line applies for erosivities derived from 5-min data for the test region and 2 yr (n = 24 770). Straight vertical and horizontal lines indicate return periods between 2 yr and 100 yr. The y axis is probability scaled, the x axis is log scaled.**

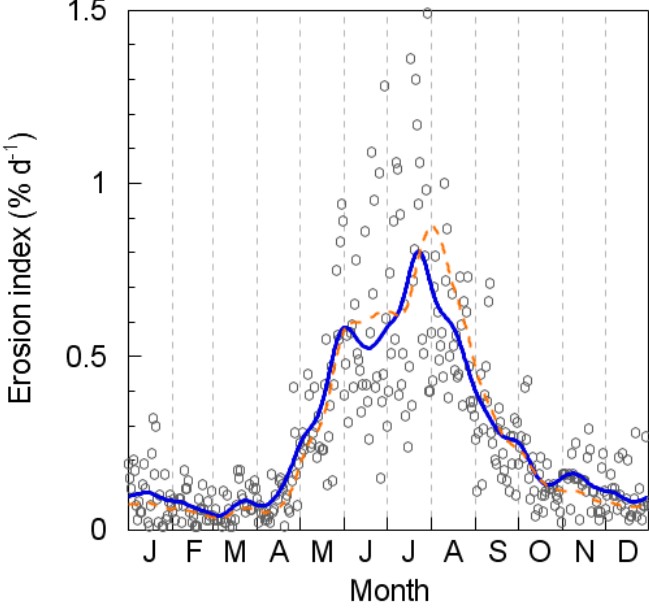

**Figure 5: Measured (circles) and smoothed (solid blue line) daily erosion index. The daily erosion index calculated from measurements between 2001 and 2016 at 115 rain gauges distributed throughout Germany is given for comparison (orange dashed line). For C-factor calculations the smoothed values can be taken from Table A2 in the Appendix.**



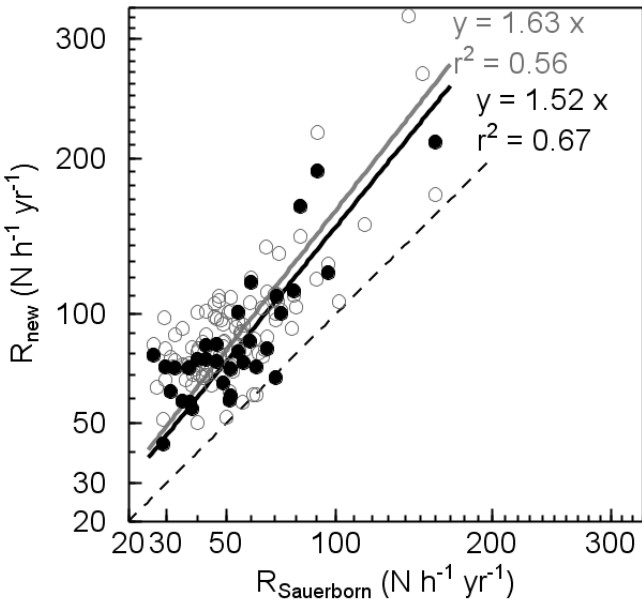

**Figure 6: Comparison of erosivities reported by Sauerborn (1994), measured mainly in the 1960s to 1980s, with recent erosivities. Recent erosivities were either determined from rain gauge measurements at the same meteorological stations (mean of 2001 to 2016; taken from Fischer et al., 2018; n = 33; black dots) or from radar data (mean of 2001 to 2017 and all radar pixels at a distance of 1.5 km from the meteorological stations; n = 101, white dots). Both axes are square-root scaled to improve resolution at low erosivities. Dashed line denotes 1:1. Solid lines are regressions through the origin.**

**Table 1: Influence of temporal resolution of rain data (5 min and 1 h), averaging (1 yr, 2 yr, and 17 yr), winsorizing and kriging on the semivariance (gamma) at three lags $h$. For complete semivariograms see Fig. 3, upper panel.**

| Variable | gamma at $h = 10$ km ($N^2$ $h^{-2}$ $yr^{-2}$) | gamma at $h = 20$ km ($N^2$ $h^{-2}$ $yr^{-2}$) | gamma at $h = 40$ km ($N^2$ $h^{-2}$ $yr^{-2}$) | Regional trend ① ($N$ $h^{-1}$ $yr^{-1}$ $km^{-1}$) | Effect of violent rain cells ② ($N$ $h^{-1}$ $yr^{-1}$ $km^{-1}$) |
|---|---|---|---|---|---|
| 5-min annual erosivity | 1925 | 2749 | 3136 | 0.2 | 2.4 |
| 5-min biannual erosivity | 1111 | 1569 | 1755 | 0.1 | 1.9 |
| 1-h annual erosivity | 1413 | 1667 | 2147 | 0.3 | 1.8 |
| 1-h biannual erosivity | 782 | 953 | 1259 | 0.2 | 1.3 |
| 1-h 17-yr mean erosivity | 144 | 197 | 315 | 0.2 | 0.5 |
| 1-h winsorized mean erosivity | 139 | 190 | 309 | 0.2 | 0.5 |
| 1-h kriged erosivity | 60 | 121 | 239 | 0.2 | 0.3 |

① The regional trend was calculated as the difference between the square roots of gamma at lags of 40 and 20 km divided by the difference in lag of 20 km.

② The effect of violent rain cells was calculated as the square root of gamma at a lag of 20 km divided by the difference in lag of 20 km minus the regional trend.





**Appendix**

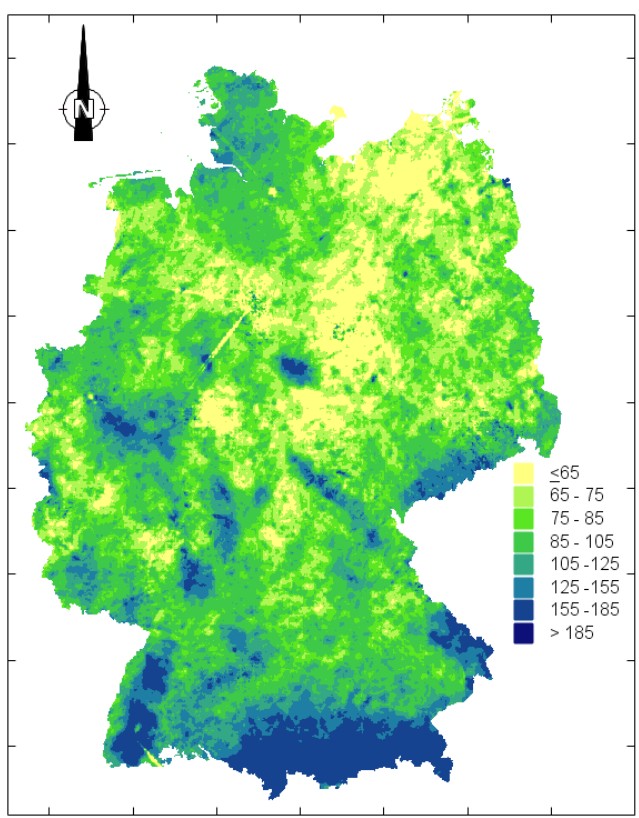

**Figure A1:** **Erosivity map of Germany from 17 yr of radar rain data before statistical smoothing by winsorizing, removal of spokes and kriging. Axes ticks represent distances of 100 km.**





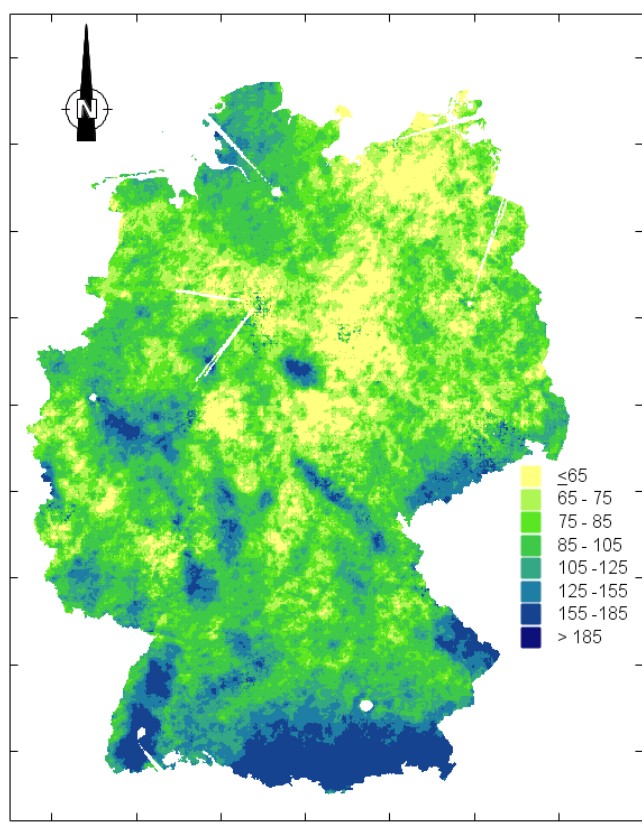

**Figure A2:   Erosivity map of Germany from 17 yr of radar rain data after winsorizing and removal of spokes but before kriging. Axes ticks represent distances of 100 km.**



**Table A1. Mean erosivity (N h$^{-1}$ yr$^{-1}$) of all German counties**

| County (Landkreis) | Identifier | Size (km²) | Mean R |
|---|---|---|---|
| Ahrweiler | 07131001 | 789 | 74.3 |
| Aichach-Friedberg | 09771111 | 780 | 109.9 |
| Alb-Donau-Kreis | 08425002 | 1358 | 103.4 |
| Altenburger Land | 16077001 | 570 | 90.5 |
| Altenkirchen (Westerwald) | 07132001 | 643 | 100.4 |
| Altmarkkreis Salzwedel | 15081026 | 2304 | 62.2 |
| Altötting | 09171111 | 569 | 128.9 |
| Alzey-Worms | 07331001 | 589 | 68.7 |
| Amberg | 09361000 | 50 | 81.0 |
| Amberg-Sulzbach | 09371111 | 1255 | 92.2 |
| Ammerland | 03451001 | 731 | 80.6 |
| Anhalt-Bitterfeld | 15082005 | 1461 | 68.0 |
| Ansbach | 09561000 | 2073 | 80.8 |
| Aschaffenburg | 09661000 | 762 | 118.7 |
| Augsburg | 09761000 | 1218 | 123.3 |
| Aurich | 03452001 | 1298 | 85.0 |
| Bad Dürkheim | 07332001 | 595 | 88.4 |
| Bad Kissingen | 09672111 | 1138 | 84.9 |
| Bad Kreuznach | 07133001 | 866 | 80.5 |
| Bad Tölz-Wolfratshausen | 09173111 | 1112 | 257.1 |
| Baden-Baden. Stadtkreis | 08211000 | 140 | 131.2 |
| Bamberg | 09461000 | 1222 | 85.5 |
| Barnim | 12060005 | 1481 | 72.7 |
| Bautzen | 14625010 | 2397 | 86.9 |
| Bayreuth | 09462000 | 1341 | 108.0 |
| Berchtesgadener Land | 09172111 | 840 | 250.0 |
| Bergstraße | 06431001 | 720 | 120.7 |
| Berlin. Stadt | 11000000 | 892 | 73.2 |
| Bernkastel-Wittlich | 07231001 | 1173 | 89.5 |
| Biberach | 08426001 | 1411 | 129.8 |
| Bielefeld. Stadt | 05711000 | 259 | 94.4 |
| Birkenfeld | 07134001 | 779 | 98.1 |
| Böblingen | 08115001 | 618 | 96.2 |
| Bochum. Stadt | 05911000 | 145 | 103.7 |
| Bodenseekreis | 08435005 | 666 | 149.3 |
| Bonn. Stadt | 05314000 | 142 | 94.5 |
| Börde | 15083020 | 2377 | 58.4 |
| Borken | 05554004 | 1426 | 94.4 |
| Bottrop. Stadt | 05512000 | 101 | 109.6 |
| Brandenburg an der Havel. Stadt | 12051000 | 229 | 78.8 |
| Braunschweig. Stadt | 03101000 | 192 | 69.3 |
| Breisgau-Hochschwarzwald | 08315003 | 1380 | 163.9 |





**Table A1. Mean erosivity (N h$^{-1}$ yr$^{-1}$) of all German counties (continued)**

| County (Landkreis) | Identifier | Size (km²) | Mean R |
|---|---|---|---|
| Bremen. Stadt | 04011000 | 326 | 78.0 |
| Bremerhaven. Stadt | 04012000 | 94 | 86.4 |
| Burgenlandkreis | 15084012 | 1419 | 77.0 |
| Calw | 08235006 | 798 | 105.6 |
| Celle | 03351001 | 1551 | 70.8 |
| Cham | 09372112 | 1527 | 106.1 |
| Chemnitz. Stadt | 14511000 | 221 | 107.5 |
| Cloppenburg | 03453001 | 1424 | 80.5 |
| Coburg | 09463000 | 639 | 82.4 |
| Cochem-Zell | 07135001 | 695 | 80.0 |
| Coesfeld | 05558004 | 1115 | 93.1 |
| Cottbus. Stadt | 12052000 | 165 | 74.0 |
| Cuxhaven | 03352002 | 2062 | 99.6 |
| Dachau | 09174111 | 580 | 115.1 |
| Dahme-Spreewald | 12061005 | 2277 | 79.2 |
| Darmstadt. Wissenschaftsstadt | 06411000 | 123 | 102.3 |
| Darmstadt-Dieburg | 06432001 | 659 | 91.5 |
| Deggendorf | 09271111 | 861 | 118.8 |
| Delmenhorst. Stadt | 03401000 | 63 | 69.6 |
| Dessau-Roßlau. Stadt | 15001000 | 246 | 65.0 |
| Diepholz | 03251001 | 1993 | 69.9 |
| Dillingen a.d.Donau | 09773111 | 792 | 94.7 |
| Dingolfing-Landau | 09279112 | 877 | 90.8 |
| Dithmarschen | 01051001 | 1444 | 122.5 |
| Donau-Ries | 09779111 | 1275 | 87.9 |
| Donnersbergkreis | 07333001 | 646 | 89.5 |
| Dortmund. Stadt | 05913000 | 280 | 88.7 |
| Dresden. Stadt | 14612000 | 328 | 96.9 |
| Duisburg. Stadt | 05112000 | 234 | 87.2 |
| Düren | 05358004 | 944 | 80.0 |
| Düsseldorf. Stadt | 05111000 | 218 | 75.6 |
| Ebersberg | 09175111 | 550 | 154.0 |
| Eichsfeld | 16061001 | 943 | 68.4 |
| Eichstätt | 09176111 | 1214 | 91.0 |
| Eifelkreis Bitburg-Prüm | 07232001 | 1634 | 82.0 |
| Eisenach. Stadt | 16056000 | 105 | 72.1 |
| Elbe-Elster | 12062024 | 1901 | 74.7 |
| Emden. Stadt | 03402000 | 112 | 73.3 |
| Emmendingen | 08316002 | 682 | 152.6 |
| Emsland | 03454001 | 2891 | 81.0 |
| Ennepe-Ruhr-Kreis | 05954004 | 412 | 116.1 |
| Enzkreis | 08236004 | 574 | 94.5 |
| Erding | 09177112 | 871 | 113.8 |





**Table A1. Mean erosivity (N h$^{-1}$ yr$^{-1}$) of all German counties (continued)**

| County (Landkreis) | Identifier | Size (km²) | Mean R |
|---|---|---|---|
| Erfurt. Stadt | 16051000 | 270 | 74.5 |
| Erlangen | 09562000 | 78 | 81.9 |
| Erlangen-Höchstadt | 09572111 | 565 | 80.2 |
| Erzgebirgskreis | 14521010 | 1827 | 136.9 |
| Essen. Stadt | 05113000 | 211 | 113.5 |
| Esslingen | 08116004 | 640 | 109.5 |
| Euskirchen | 05366004 | 1255 | 82.3 |
| Flensburg. Stadt | 01001000 | 57 | 108.8 |
| Forchheim | 09474119 | 643 | 98.3 |
| Frankenthal (Pfalz). kreisfreie Stadt | 07311000 | 44 | 80.3 |
| Frankfurt (Oder). Stadt | 12053000 | 148 | 89.6 |
| Frankfurt am Main. Stadt | 06412000 | 249 | 94.2 |
| Freiburg im Breisgau. Stadtkreis | 08311000 | 155 | 139.1 |
| Freising | 09178113 | 798 | 107.1 |
| Freudenstadt | 08237002 | 873 | 160.4 |
| Freyung-Grafenau | 09272116 | 985 | 175.0 |
| Friesland | 03455007 | 619 | 85.5 |
| Fulda | 06631001 | 1382 | 85.6 |
| Fürstenfeldbruck | 09179111 | 435 | 133.2 |
| Fürth | 09563000 | 371 | 78.3 |
| Garmisch-Partenkirchen | 09180112 | 1012 | 215.8 |
| Gelsenkirchen. Stadt | 05513000 | 106 | 108.9 |
| Gera. Stadt | 16052000 | 152 | 78.2 |
| Germersheim | 07334001 | 464 | 90.3 |
| Gießen | 06531001 | 857 | 88.0 |
| Gifhorn | 03151001 | 1570 | 72.1 |
| Göppingen | 08117001 | 643 | 119.3 |
| Görlitz | 14626010 | 2113 | 96.8 |
| Goslar | 03153002 | 969 | 122.9 |
| Gotha | 16067003 | 936 | 82.2 |
| Göttingen | 03159001 | 1756 | 92.2 |
| Grafschaft Bentheim | 03456001 | 985 | 80.6 |
| Greiz | 16076003 | 846 | 84.2 |
| Groß-Gerau | 06433001 | 454 | 73.4 |
| Günzburg | 09774111 | 764 | 112.4 |
| Gütersloh | 05754004 | 971 | 79.6 |
| Hagen. Stadt der FernUniversität | 05914000 | 161 | 104.8 |
| Halle (Saale). Stadt | 15002000 | 136 | 78.3 |
| Hamburg. Freie und Hansestadt | 02000000 | 753 | 87.7 |
| Hameln-Pyrmont | 03252001 | 799 | 79.0 |
| Hamm. Stadt | 05915000 | 228 | 77.6 |
| Harburg | 03353001 | 1250 | 88.4 |
| Harz | 15085040 | 2108 | 73.0 |



**Table A1. Mean erosivity (N h$^{-1}$ yr$^{-1}$) of all German counties (continued)**

| County (Landkreis) | Identifier | Size (km²) | Mean R |
|---|---|---|---|
| Haßberge | 09674111 | 957 | 83.4 |
| Havelland | 12063036 | 1728 | 74.5 |
| Heidekreis | 03358001 | 1883 | 80.3 |
| Heidelberg. Stadtkreis | 08221000 | 109 | 124.1 |
| Heidenheim | 08135010 | 628 | 99.8 |
| Heilbronn | 08125001 | 1100 | 90.5 |
| Heilbronn. Stadtkreis | 08121000 | 101 | 79.9 |
| Heinsberg | 05370004 | 630 | 71.1 |
| Helmstedt | 03154001 | 676 | 61.0 |
| Herford | 05758004 | 451 | 89.9 |
| Herne. Stadt | 05916000 | 52 | 104.6 |
| Hersfeld-Rotenburg | 06632001 | 1099 | 76.6 |
| Herzogtum Lauenburg | 01053001 | 1263 | 78.6 |
| Hildburghausen | 16069001 | 938 | 90.1 |
| Hildesheim | 03254002 | 1208 | 74.4 |
| Hochsauerlandkreis | 05958004 | 1963 | 107.5 |
| Hochtaunuskreis | 06434001 | 482 | 108.3 |
| Hof | 09464000 | 952 | 95.5 |
| Hohenlohekreis | 08126011 | 778 | 97.5 |
| Holzminden | 03255001 | 695 | 84.1 |
| Höxter | 05762004 | 1202 | 80.5 |
| Ilm-Kreis | 16070001 | 844 | 97.1 |
| Ingolstadt | 09161000 | 134 | 90.5 |
| Jena. Stadt | 16053000 | 115 | 79.0 |
| Jerichower Land | 15086005 | 1589 | 69.9 |
| Kaiserslautern | 07335002 | 642 | 97.5 |
| Kaiserslautern. kreisfreie Stadt | 07312000 | 141 | 100.7 |
| Karlsruhe | 08215007 | 1086 | 90.8 |
| Karlsruhe. Stadtkreis | 08212000 | 174 | 98.7 |
| Kassel | 06633001 | 1296 | 69.7 |
| Kassel. documenta-Stadt | 06611000 | 105 | 66.7 |
| Kaufbeuren | 09762000 | 40 | 168.0 |
| Kelheim | 09273111 | 1065 | 91.9 |
| Kempten (Allgäu) | 09763000 | 63 | 222.1 |
| Kiel. Landeshauptstadt | 01002000 | 120 | 92.5 |
| Kitzingen | 09675111 | 684 | 81.1 |
| Kleve | 05154004 | 1238 | 98.9 |
| Koblenz. kreisfreie Stadt | 07111000 | 106 | 80.2 |
| Köln. Stadt | 05315000 | 408 | 91.3 |
| Konstanz | 08335001 | 819 | 121.1 |
| Krefeld. Stadt | 05114000 | 137 | 83.1 |
| Kronach | 09476145 | 652 | 107.4 |
| Kulmbach | 09477117 | 658 | 100.4 |





**Table A1. Mean erosivity (N h$^{-1}$ yr$^{-1}$) of all German counties (continued)**

| County (Landkreis) | Identifier | Size (km²) | Mean R |
|---|---|---|---|
| Kusel | 07336001 | 575 | 101.2 |
| Kyffhäuserkreis | 16065001 | 1038 | 61.5 |
| Lahn-Dill-Kreis | 06532001 | 1067 | 102.7 |
| Landau in der Pfalz. kreisfreie Stadt | 07313000 | 83 | 104.7 |
| Landkreis Rostock | 13072001 | 3429 | 59.5 |
| Landsberg am Lech | 09181111 | 804 | 156.0 |
| Landshut | 09261000 | 1414 | 103.9 |
| Leer | 03457002 | 1089 | 72.8 |
| Leipzig | 14729010 | 1652 | 79.9 |
| Leipzig. Stadt | 14713000 | 299 | 87.6 |
| Leverkusen. Stadt | 05316000 | 79 | 98.0 |
| Lichtenfels | 09478111 | 520 | 80.7 |
| Limburg-Weilburg | 06533001 | 740 | 95.0 |
| Lindau (Bodensee) | 09776111 | 323 | 306.6 |
| Lippe | 05766004 | 1247 | 99.9 |
| Lörrach | 08336004 | 809 | 182.7 |
| Lübeck. Hansestadt | 01003000 | 212 | 76.1 |
| Lüchow-Dannenberg | 03354001 | 1227 | 73.6 |
| Ludwigsburg | 08118001 | 687 | 88.6 |
| Ludwigshafen am Rhein. kreisfreie Stadt | 07314000 | 78 | 87.5 |
| Ludwigslust-Parchim | 13076001 | 4768 | 71.8 |
| Lüneburg | 03355001 | 1327 | 80.4 |
| Magdeburg. Landeshauptstadt | 15003000 | 201 | 54.2 |
| Main-Kinzig-Kreis | 06435001 | 1398 | 110.4 |
| Main-Spessart | 09677114 | 1323 | 95.1 |
| Main-Tauber-Kreis | 08128006 | 1306 | 93.8 |
| Main-Taunus-Kreis | 06436001 | 222 | 102.2 |
| Mainz. kreisfreie Stadt | 07315000 | 98 | 68.4 |
| Mainz-Bingen | 07339001 | 607 | 68.8 |
| Mannheim. Stadtkreis | 08222000 | 145 | 92.9 |
| Mansfeld-Südharz | 15087010 | 1456 | 66.8 |
| Marburg-Biedenkopf | 06534001 | 1264 | 83.0 |
| Märkischer Kreis | 05962004 | 1064 | 121.3 |
| Märkisch-Oderland | 12064009 | 2159 | 80.6 |
| Mayen-Koblenz | 07137001 | 819 | 69.6 |
| Mecklenburgische Seenplatte | 13071001 | 5496 | 67.2 |
| Meißen | 14627010 | 1458 | 76.5 |
| Memmingen | 09764000 | 70 | 157.0 |
| Merzig-Wadern | 10042111 | 559 | 108.4 |
| Mettmann | 05158004 | 409 | 101.9 |
| Miesbach | 09182111 | 867 | 281.4 |
| Miltenberg | 09676111 | 716 | 105.2 |
| Minden-Lübbecke | 05770004 | 1153 | 77.0 |





**Table A1. Mean erosivity (N h$^{-1}$ yr$^{-1}$) of all German counties (continued)**

| County (Landkreis) | Identifier | Size (km²) | Mean R |
|---|---|---|---|
| Mittelsachsen | 14522010 | 2115 | 102.9 |
| Mönchengladbach. Stadt | 05116000 | 171 | 76.8 |
| Mühldorf a.Inn | 09183112 | 805 | 110.8 |
| Mülheim an der Ruhr. Stadt | 05117000 | 92 | 101.4 |
| München | 09184112 | 664 | 161.1 |
| München. Landeshauptstadt | 09162000 | 311 | 149.4 |
| Münster. Stadt | 05515000 | 304 | 88.9 |
| Neckar-Odenwald-Kreis | 08225001 | 1127 | 104.4 |
| Neuburg-Schrobenhausen | 09185113 | 740 | 93.8 |
| Neumarkt i.d.OPf. | 09373112 | 1345 | 92.8 |
| Neumünster. Stadt | 01004000 | 71 | 100.2 |
| Neunkirchen | 10043111 | 249 | 117.5 |
| Neustadt a.d.Aisch-Bad Windsheim | 09575112 | 1268 | 81.8 |
| Neustadt a.d.Waldnaab | 09374111 | 1428 | 88.8 |
| Neustadt an der Weinstraße. kreisfreie Stadt | 07316000 | 118 | 92.1 |
| Neu-Ulm | 09775111 | 516 | 121.0 |
| Neuwied | 07138002 | 629 | 80.4 |
| Nienburg (Weser) | 03256001 | 1403 | 66.6 |
| Nordfriesland | 01054001 | 2090 | 101.3 |
| Nordhausen | 16062002 | 714 | 63.2 |
| Nordsachsen | 14730010 | 2028 | 73.1 |
| Nordwestmecklenburg | 13074001 | 2125 | 68.6 |
| Northeim | 03155001 | 1270 | 81.7 |
| Nürnberg | 09564000 | 188 | 82.7 |
| Nürnberger Land | 09574111 | 798 | 103.9 |
| Oberallgäu | 09780112 | 1529 | 315.6 |
| Oberbergischer Kreis | 05374004 | 920 | 144.4 |
| Oberhausen. Stadt | 05119000 | 78 | 103.9 |
| Oberhavel | 12065036 | 1808 | 71.3 |
| Oberspreewald-Lausitz | 12066008 | 1224 | 75.4 |
| Odenwaldkreis | 06437001 | 626 | 124.6 |
| Oder-Spree | 12067024 | 2259 | 81.0 |
| Offenbach | 06438001 | 357 | 83.0 |
| Offenbach am Main. Stadt | 06413000 | 45 | 88.5 |
| Oldenburg | 03458001 | 1067 | 73.1 |
| Oldenburg (Oldenburg). Stadt | 03403000 | 104 | 78.8 |
| Olpe | 05966004 | 713 | 124.0 |
| Ortenaukreis | 08317001 | 1864 | 137.3 |
| Osnabrück | 03459001 | 2125 | 83.1 |
| Osnabrück. Stadt | 03404000 | 120 | 85.0 |
| Ostalbkreis | 08136002 | 1511 | 106.0 |
| Ostallgäu | 09777111 | 1394 | 215.5 |
| Osterholz | 03356001 | 654 | 84.5 |





**Table A1. Mean erosivity (N h$^{-1}$ yr$^{-1}$) of all German counties (continued)**

| County (Landkreis) | Identifier | Size (km²) | Mean R |
|---|---|---|---|
| Ostholstein | 01055001 | 1394 | 77.3 |
| Ostprignitz-Ruppin | 12068052 | 2526 | 77.0 |
| Paderborn | 05774004 | 1248 | 91.8 |
| Passau | 09262000 | 1600 | 121.0 |
| Peine | 03157001 | 536 | 65.8 |
| Pfaffenhofen a.d.Ilm | 09186113 | 761 | 100.8 |
| Pforzheim. Stadtkreis | 08231000 | 98 | 89.3 |
| Pinneberg | 01056001 | 664 | 97.6 |
| Pirmasens. kreisfreie Stadt | 07317000 | 62 | 99.4 |
| Plön | 01057001 | 1084 | 83.3 |
| Potsdam. Stadt | 12054000 | 187 | 70.5 |
| Potsdam-Mittelmark | 12069017 | 2593 | 78.3 |
| Prignitz | 12070008 | 2139 | 69.6 |
| Rastatt | 08216002 | 740 | 126.3 |
| Ravensburg | 08436001 | 1633 | 178.3 |
| Recklinghausen | 05562004 | 763 | 100.3 |
| Regen | 09276111 | 975 | 164.4 |
| Regensburg | 09362000 | 1473 | 84.4 |
| Region Hannover | 03241001 | 2299 | 70.8 |
| Regionalverband Saarbrücken | 10041100 | 413 | 103.7 |
| Remscheid. Stadt | 05120000 | 74 | 150.4 |
| Rems-Murr-Kreis | 08119001 | 858 | 119.4 |
| Rendsburg-Eckernförde | 01058001 | 2190 | 104.7 |
| Reutlingen | 08415014 | 1093 | 119.8 |
| Rhein-Erft-Kreis | 05362004 | 705 | 76.4 |
| Rheingau-Taunus-Kreis | 06439001 | 814 | 88.0 |
| Rhein-Hunsrück-Kreis | 07140001 | 994 | 84.1 |
| Rheinisch-Bergischer Kreis | 05378004 | 439 | 120.3 |
| Rhein-Kreis Neuss | 05162004 | 579 | 72.2 |
| Rhein-Lahn-Kreis | 07141001 | 783 | 87.8 |
| Rhein-Neckar-Kreis | 08226003 | 1062 | 114.3 |
| Rhein-Pfalz-Kreis | 07338001 | 305 | 88.5 |
| Rhein-Sieg-Kreis | 05382004 | 1155 | 96.8 |
| Rhön-Grabfeld | 09673113 | 1022 | 74.7 |
| Rosenheim | 09163000 | 1477 | 210.4 |
| Rostock | 13003000 | 181 | 68.4 |
| Rotenburg (Wümme) | 03357001 | 2075 | 92.6 |
| Roth | 09576111 | 895 | 84.9 |
| Rottal-Inn | 09277111 | 1281 | 102.6 |
| Rottweil | 08325001 | 771 | 115.4 |
| Saale-Holzland-Kreis | 16074001 | 816 | 85.5 |
| Saalekreis | 15088020 | 1440 | 73.1 |
| Saale-Orla-Kreis | 16075002 | 1152 | 84.8 |





**Table A1. Mean erosivity (N h⁻¹ yr⁻¹) of all German counties (continued)**

| County (Landkreis) | Identifier | Size (km²) | Mean R |
|---|---|---|---|
| Saalfeld-Rudolstadt | 16073001 | 1036 | 87.1 |
| Saarlouis | 10044111 | 461 | 105.3 |
| Saarpfalz-Kreis | 10045111 | 420 | 101.3 |
| Sächsische Schweiz-Osterzgebirge | 14628010 | 1654 | 111.7 |
| Salzgitter. Stadt | 03102000 | 225 | 73.6 |
| Salzlandkreis | 15089005 | 1435 | 61.5 |
| Schaumburg | 03257001 | 677 | 81.9 |
| Schleswig-Flensburg | 01059001 | 2072 | 106.8 |
| Schmalkalden-Meiningen | 16066001 | 1211 | 84.7 |
| Schwabach | 09565000 | 41 | 73.6 |
| Schwäbisch Hall | 08127008 | 1485 | 94.6 |
| Schwalm-Eder-Kreis | 06634001 | 1541 | 70.7 |
| Schwandorf | 09376112 | 1458 | 81.5 |
| Schwarzwald-Baar-Kreis | 08326003 | 1028 | 135.6 |
| Schweinfurt | 09662000 | 877 | 71.0 |
| Schwerin | 13004000 | 130 | 64.8 |
| Segeberg | 01060002 | 1346 | 92.6 |
| Siegen-Wittgenstein | 05970004 | 1136 | 121.3 |
| Sigmaringen | 08437005 | 1206 | 118.3 |
| Soest | 05974004 | 1332 | 88.1 |
| Solingen. Klingenstadt | 05122000 | 89 | 115.1 |
| Sömmerda | 16068001 | 807 | 64.6 |
| Sonneberg | 16072001 | 433 | 125.5 |
| Speyer. kreisfreie Stadt | 07318000 | 43 | 89.0 |
| Spree-Neiße | 12071028 | 1658 | 77.5 |
| St. Wendel | 10046111 | 478 | 122.4 |
| Stade | 03359001 | 1268 | 97.1 |
| Städteregion Aachen | 05334002 | 707 | 105.9 |
| Starnberg | 09188113 | 487 | 169.0 |
| Steinburg | 01061001 | 1057 | 107.7 |
| Steinfurt | 05566004 | 1800 | 95.6 |
| Stendal | 15090003 | 2437 | 64.5 |
| Stormarn | 01062001 | 766 | 87.6 |
| Straubing | 09263000 | 67 | 90.5 |
| Straubing-Bogen | 09278112 | 1201 | 103.8 |
| Stuttgart. Stadtkreis | 08111000 | 210 | 92.0 |
| Südliche Weinstraße | 07337001 | 641 | 104.2 |
| Südwestpfalz | 07340001 | 957 | 109.7 |
| Suhl. Stadt | 16054000 | 102 | 123.8 |
| Teltow-Fläming | 12072002 | 2104 | 73.0 |
| Tirschenreuth | 09377112 | 1085 | 96.5 |
| Traunstein | 09189111 | 1533 | 232.0 |
| Trier. kreisfreie Stadt | 07211000 | 116 | 77.3 |



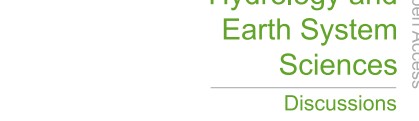

**Table A1. Mean erosivity (N h$^{-1}$ yr$^{-1}$) of all German counties (continued)**

| County (Landkreis) | Identifier | Size (km²) | Mean R |
| --- | --- | --- | --- |
| Trier-Saarburg | 07235001 | 1109 | 92.8 |
| Tübingen | 08416006 | 521 | 114.2 |
| Tuttlingen | 08327002 | 735 | 124.3 |
| Uckermark | 12073008 | 3077 | 74.2 |
| Uelzen | 03360001 | 1463 | 79.2 |
| Ulm. Stadtkreis | 08421000 | 119 | 93.7 |
| Unna | 05978004 | 544 | 83.6 |
| Unstrut-Hainich-Kreis | 16064001 | 979 | 63.2 |
| Unterallgäu | 09778111 | 1230 | 161.8 |
| Vechta | 03460001 | 815 | 73.2 |
| Verden | 03361001 | 790 | 76.5 |
| Viersen | 05166004 | 566 | 78.0 |
| Vogelsbergkreis | 06535001 | 1460 | 95.3 |
| Vogtlandkreis | 14523010 | 1412 | 101.5 |
| Vorpommern-Greifswald | 13075001 | 3953 | 72.1 |
| Vorpommern-Rügen | 13073001 | 3213 | 66.2 |
| Vulkaneifel | 07233002 | 915 | 88.4 |
| Waldeck-Frankenberg | 06635001 | 1850 | 72.8 |
| Waldshut | 08337002 | 1133 | 166.8 |
| Warendorf | 05570004 | 1321 | 75.6 |
| Wartburgkreis | 16063003 | 1307 | 75.7 |
| Weiden i.d.OPf. | 09363000 | 71 | 91.7 |
| Weilheim-Schongau | 09190111 | 968 | 214.4 |
| Weimar. Stadt | 16055000 | 84 | 70.6 |
| Weimarer Land | 16071001 | 804 | 73.8 |
| Weißenburg-Gunzenhausen | 09577111 | 971 | 94.1 |
| Werra-Meißner-Kreis | 06636001 | 1025 | 73.9 |
| Wesel | 05170004 | 1046 | 92.7 |
| Wesermarsch | 03461001 | 830 | 77.9 |
| Westerwaldkreis | 07143001 | 992 | 100.5 |
| Wetteraukreis | 06440001 | 1102 | 97.0 |
| Wiesbaden. Landeshauptstadt | 06414000 | 204 | 79.4 |
| Wilhelmshaven. Stadt | 03405000 | 108 | 94.8 |
| Wittenberg | 15091010 | 1943 | 77.8 |
| Wittmund | 03462001 | 661 | 96.6 |
| Wolfenbüttel | 03158002 | 724 | 71.7 |
| Wolfsburg. Stadt | 03103000 | 205 | 66.3 |
| Worms. kreisfreie Stadt | 07319000 | 109 | 71.7 |
| Wunsiedel i.Fichtelgebirge | 09479111 | 606 | 92.9 |
| Wuppertal. Stadt | 05124000 | 169 | 128.8 |
| Würzburg | 09663000 | 1055 | 85.0 |
| Zollernalbkreis | 08417002 | 918 | 118.0 |





**Table A1. Mean erosivity (N h$^{-1}$ yr$^{-1}$) of all German counties (continued)**

| County (Landkreis) | Identifier | Size (km²) | Mean R |
|---|---|---|---|
| Zweibrücken. kreisfreie Stadt | 07320000 | 71 | 90.4 |
| Zwickau | 14524010 | 950 | 102.4 |

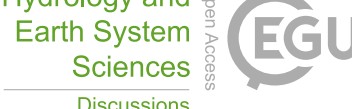

**Table A2. Daily erosion index**

| Date | Daily erosivity (%) | Erosivity since 1 Jan (%) | Date | Daily erosivity (%) | Erosivity since 1 Jan (%) | Date | Daily erosivity (%) | Erosivity since 1 Jan (%) | Date | Daily erosivity (%) | Erosivity since 1 Jan (%) |
|---|---|---|---|---|---|---|---|---|---|---|---|
| 1 Jan | 0.09 | 0.1 | 1 Apr | 0.07 | 6.9 | 1 Jul | 0.58 | 39.3 | 1 Oct | 0.25 | 87.8 |
| 2 Jan | 0.10 | 0.2 | 2 Apr | 0.07 | 7.0 | 2 Jul | 0.58 | 39.9 | 2 Oct | 0.25 | 88.0 |
| 3 Jan | 0.10 | 0.3 | 3 Apr | 0.07 | 7.1 | 3 Jul | 0.59 | 40.5 | 3 Oct | 0.25 | 88.3 |
| 4 Jan | 0.10 | 0.4 | 4 Apr | 0.07 | 7.1 | 4 Jul | 0.59 | 41.1 | 4 Oct | 0.24 | 88.5 |
| 5 Jan | 0.10 | 0.5 | 5 Apr | 0.07 | 7.2 | 5 Jul | 0.60 | 41.7 | 5 Oct | 0.23 | 88.8 |
| 6 Jan | 0.10 | 0.6 | 6 Apr | 0.07 | 7.3 | 6 Jul | 0.60 | 42.3 | 6 Oct | 0.23 | 89.0 |
| 7 Jan | 0.10 | 0.7 | 7 Apr | 0.07 | 7.4 | 7 Jul | 0.61 | 42.9 | 7 Oct | 0.22 | 89.2 |
| 8 Jan | 0.10 | 0.8 | 8 Apr | 0.07 | 7.4 | 8 Jul | 0.61 | 43.5 | 8 Oct | 0.21 | 89.4 |
| 9 Jan | 0.10 | 0.9 | 9 Apr | 0.07 | 7.5 | 9 Jul | 0.62 | 44.1 | 9 Oct | 0.20 | 89.6 |
| 10 Jan | 0.10 | 1.0 | 10 Apr | 0.07 | 7.6 | 10 Jul | 0.63 | 44.8 | 10 Oct | 0.19 | 89.8 |
| 11 Jan | 0.10 | 1.1 | 11 Apr | 0.08 | 7.6 | 11 Jul | 0.64 | 45.4 | 11 Oct | 0.18 | 90.0 |
| 12 Jan | 0.11 | 1.2 | 12 Apr | 0.08 | 7.7 | 12 Jul | 0.65 | 46.1 | 12 Oct | 0.17 | 90.2 |
| 13 Jan | 0.11 | 1.3 | 13 Apr | 0.09 | 7.8 | 13 Jul | 0.67 | 46.7 | 13 Oct | 0.17 | 90.3 |
| 14 Jan | 0.11 | 1.4 | 14 Apr | 0.09 | 7.9 | 14 Jul | 0.68 | 47.4 | 14 Oct | 0.16 | 90.5 |
| 15 Jan | 0.11 | 1.5 | 15 Apr | 0.10 | 8.0 | 15 Jul | 0.70 | 48.1 | 15 Oct | 0.15 | 90.6 |
| 16 Jan | 0.11 | 1.6 | 16 Apr | 0.10 | 8.1 | 16 Jul | 0.71 | 48.8 | 16 Oct | 0.15 | 90.8 |
| 17 Jan | 0.11 | 1.7 | 17 Apr | 0.11 | 8.2 | 17 Jul | 0.73 | 49.5 | 17 Oct | 0.14 | 90.9 |
| 18 Jan | 0.10 | 1.8 | 18 Apr | 0.11 | 8.3 | 18 Jul | 0.75 | 50.3 | 18 Oct | 0.14 | 91.1 |
| 19 Jan | 0.10 | 1.9 | 19 Apr | 0.12 | 8.4 | 19 Jul | 0.76 | 51.1 | 19 Oct | 0.13 | 91.2 |
| 20 Jan | 0.10 | 2.0 | 20 Apr | 0.13 | 8.6 | 20 Jul | 0.78 | 51.8 | 20 Oct | 0.13 | 91.3 |
| 21 Jan | 0.10 | 2.1 | 21 Apr | 0.14 | 8.7 | 21 Jul | 0.79 | 52.6 | 21 Oct | 0.13 | 91.5 |
| 22 Jan | 0.10 | 2.2 | 22 Apr | 0.15 | 8.9 | 22 Jul | 0.80 | 53.4 | 22 Oct | 0.13 | 91.6 |
| 23 Jan | 0.10 | 2.3 | 23 Apr | 0.15 | 9.0 | 23 Jul | 0.80 | 54.2 | 23 Oct | 0.13 | 91.7 |
| 24 Jan | 0.09 | 2.4 | 24 Apr | 0.16 | 9.2 | 24 Jul | 0.80 | 55.0 | 24 Oct | 0.13 | 91.9 |
| 25 Jan | 0.09 | 2.5 | 25 Apr | 0.17 | 9.3 | 25 Jul | 0.80 | 55.8 | 25 Oct | 0.13 | 92.0 |
| 26 Jan | 0.09 | 2.6 | 26 Apr | 0.18 | 9.5 | 26 Jul | 0.79 | 56.6 | 26 Oct | 0.13 | 92.1 |
| 27 Jan | 0.09 | 2.7 | 27 Apr | 0.20 | 9.7 | 27 Jul | 0.79 | 57.4 | 27 Oct | 0.13 | 92.2 |
| 28 Jan | 0.09 | 2.8 | 28 Apr | 0.21 | 9.9 | 28 Jul | 0.77 | 58.2 | 28 Oct | 0.14 | 92.4 |
| 29 Jan | 0.09 | 2.9 | 29 Apr | 0.22 | 10.2 | 29 Jul | 0.76 | 58.9 | 29 Oct | 0.14 | 92.5 |
| 30 Jan | 0.08 | 3.0 | 30 Apr | 0.23 | 10.4 | 30 Jul | 0.74 | 59.7 | 30 Oct | 0.14 | 92.7 |
| 31 Jan | 0.08 | 3.0 | | | | 31 Jul | 0.73 | 60.4 | 31 Oct | 0.14 | 92.8 |





**Table A2. Daily erosion index (continued)**

| Date | Daily erosivity (%) | Erosivity since 1 Jan (%) | Date | Daily erosivity (%) | Erosivity since 1 Jan (%) | Date | Daily erosivity (%) | Erosivity since 1 Jan (%) | Date | Daily erosivity (%) | Erosivity since 1 Jan (%) |
|---|---|---|---|---|---|---|---|---|---|---|---|
| 1 Feb | 0.08 | 3.1 | 1 May | 0.24 | 10.6 | 1 Aug | 0.71 | 61.1 | 1 Nov | 0.15 | 93.0 |
| 2 Feb | 0.08 | 3.2 | 2 May | 0.25 | 10.9 | 2 Aug | 0.70 | 61.8 | 2 Nov | 0.15 | 93.1 |
| 3 Feb | 0.08 | 3.3 | 3 May | 0.26 | 11.1 | 3 Aug | 0.68 | 62.5 | 3 Nov | 0.15 | 93.3 |
| 4 Feb | 0.08 | 3.4 | 4 May | 0.26 | 11.4 | 4 Aug | 0.67 | 63.2 | 4 Nov | 0.16 | 93.4 |
| 5 Feb | 0.08 | 3.5 | 5 May | 0.27 | 11.7 | 5 Aug | 0.66 | 63.8 | 5 Nov | 0.16 | 93.6 |
| 6 Feb | 0.08 | 3.5 | 6 May | 0.28 | 11.9 | 6 Aug | 0.65 | 64.5 | 6 Nov | 0.16 | 93.7 |
| 7 Feb | 0.08 | 3.6 | 7 May | 0.28 | 12.2 | 7 Aug | 0.64 | 65.1 | 7 Nov | 0.16 | 93.9 |
| 8 Feb | 0.08 | 3.7 | 8 May | 0.29 | 12.5 | 8 Aug | 0.63 | 65.8 | 8 Nov | 0.16 | 94.1 |
| 9 Feb | 0.08 | 3.8 | 9 May | 0.29 | 12.8 | 9 Aug | 0.63 | 66.4 | 9 Nov | 0.16 | 94.2 |
| 10 Feb | 0.08 | 3.8 | 10 May | 0.30 | 13.1 | 10 Aug | 0.62 | 67.0 | 10 Nov | 0.16 | 94.4 |
| 11 Feb | 0.07 | 3.9 | 11 May | 0.30 | 13.4 | 11 Aug | 0.62 | 67.6 | 11 Nov | 0.16 | 94.5 |
| 12 Feb | 0.07 | 4.0 | 12 May | 0.31 | 13.7 | 12 Aug | 0.61 | 68.2 | 12 Nov | 0.16 | 94.7 |
| 13 Feb | 0.07 | 4.1 | 13 May | 0.32 | 14.0 | 13 Aug | 0.61 | 68.9 | 13 Nov | 0.16 | 94.9 |
| 14 Feb | 0.07 | 4.1 | 14 May | 0.33 | 14.4 | 14 Aug | 0.60 | 69.5 | 14 Nov | 0.15 | 95.0 |
| 15 Feb | 0.07 | 4.2 | 15 May | 0.34 | 14.7 | 15 Aug | 0.60 | 70.1 | 15 Nov | 0.15 | 95.2 |
| 16 Feb | 0.07 | 4.3 | 16 May | 0.35 | 15.0 | 16 Aug | 0.59 | 70.6 | 16 Nov | 0.15 | 95.3 |
| 17 Feb | 0.06 | 4.3 | 17 May | 0.36 | 15.4 | 17 Aug | 0.59 | 71.2 | 17 Nov | 0.15 | 95.5 |
| 18 Feb | 0.06 | 4.4 | 18 May | 0.37 | 15.8 | 18 Aug | 0.58 | 71.8 | 18 Nov | 0.14 | 95.6 |
| 19 Feb | 0.06 | 4.5 | 19 May | 0.39 | 16.2 | 19 Aug | 0.57 | 72.4 | 19 Nov | 0.14 | 95.7 |
| 20 Feb | 0.06 | 4.5 | 20 May | 0.41 | 16.6 | 20 Aug | 0.56 | 72.9 | 20 Nov | 0.13 | 95.9 |
| 21 Feb | 0.06 | 4.6 | 21 May | 0.43 | 17.0 | 21 Aug | 0.55 | 73.5 | 21 Nov | 0.13 | 96.0 |
| 22 Feb | 0.06 | 4.6 | 22 May | 0.45 | 17.4 | 22 Aug | 0.54 | 74.0 | 22 Nov | 0.13 | 96.1 |
| 23 Feb | 0.05 | 4.7 | 23 May | 0.47 | 17.9 | 23 Aug | 0.52 | 74.5 | 23 Nov | 0.13 | 96.3 |
| 24 Feb | 0.05 | 4.7 | 24 May | 0.48 | 18.4 | 24 Aug | 0.51 | 75.1 | 24 Nov | 0.12 | 96.4 |
| 25 Feb | 0.05 | 4.8 | 25 May | 0.50 | 18.9 | 25 Aug | 0.50 | 75.6 | 25 Nov | 0.12 | 96.5 |
| 26 Feb | 0.05 | 4.8 | 26 May | 0.52 | 19.4 | 26 Aug | 0.48 | 76.0 | 26 Nov | 0.12 | 96.6 |
| 27 Feb | 0.05 | 4.9 | 27 May | 0.53 | 19.9 | 27 Aug | 0.47 | 76.5 | 27 Nov | 0.12 | 96.7 |
| 28 Feb | 0.05 | 4.9 | 28 May | 0.55 | 20.5 | 28 Aug | 0.46 | 77.0 | 28 Nov | 0.12 | 96.9 |
|  |  |  | 29 May | 0.56 | 21.1 | 29 Aug | 0.44 | 77.4 | 29 Nov | 0.11 | 97.0 |
|  |  |  | 30 May | 0.57 | 21.6 | 30 Aug | 0.43 | 77.8 | 30 Nov | 0.11 | 97.1 |
|  |  |  | 31 May | 0.58 | 22.2 | 31 Aug | 0.42 | 78.3 |  |  |  |





**Table A2. Daily erosion index (continued)**

| Date | Daily erosivity (%) | Erosivity since 1 Jan (%) | Date | Daily erosivity (%) | Erosivity since 1 Jan (%) | Date | Daily erosivity (%) | Erosivity since 1 Jan (%) | Date | Daily erosivity (%) | Erosivity since 1 Jan (%) |
|---|---|---|---|---|---|---|---|---|---|---|---|
| 1 Mar | 0.04 | 5.0 | 1 Jun | 0.58 | 22.8 | 1 Sep | 0.41 | 78.7 | 1 Dec | 0.11 | 97.2 |
| 2 Mar | 0.04 | 5.0 | 2 Jun | 0.58 | 23.4 | 2 Sep | 0.40 | 79.1 | 2 Dec | 0.11 | 97.3 |
| 3 Mar | 0.04 | 5.1 | 3 Jun | 0.58 | 24.0 | 3 Sep | 0.39 | 79.4 | 3 Dec | 0.11 | 97.4 |
| 4 Mar | 0.04 | 5.1 | 4 Jun | 0.58 | 24.5 | 4 Sep | 0.38 | 79.8 | 4 Dec | 0.11 | 97.5 |
| 5 Mar | 0.04 | 5.1 | 5 Jun | 0.58 | 25.1 | 5 Sep | 0.37 | 80.2 | 5 Dec | 0.11 | 97.6 |
| 6 Mar | 0.04 | 5.2 | 6 Jun | 0.58 | 25.7 | 6 Sep | 0.36 | 80.6 | 6 Dec | 0.11 | 97.7 |
| 7 Mar | 0.04 | 5.2 | 7 Jun | 0.57 | 26.3 | 7 Sep | 0.35 | 80.9 | 7 Dec | 0.11 | 97.9 |
| 8 Mar | 0.04 | 5.3 | 8 Jun | 0.57 | 26.8 | 8 Sep | 0.35 | 81.3 | 8 Dec | 0.11 | 98.0 |
| 9 Mar | 0.04 | 5.3 | 9 Jun | 0.56 | 27.4 | 9 Sep | 0.34 | 81.6 | 9 Dec | 0.10 | 98.1 |
| 10 Mar | 0.05 | 5.4 | 10 Jun | 0.55 | 27.9 | 10 Sep | 0.33 | 81.9 | 10 Dec | 0.10 | 98.2 |
| 11 Mar | 0.05 | 5.4 | 11 Jun | 0.55 | 28.5 | 11 Sep | 0.33 | 82.3 | 11 Dec | 0.10 | 98.3 |
| 12 Mar | 0.05 | 5.5 | 12 Jun | 0.54 | 29.0 | 12 Sep | 0.32 | 82.6 | 12 Dec | 0.10 | 98.4 |
| 13 Mar | 0.05 | 5.5 | 13 Jun | 0.54 | 29.6 | 13 Sep | 0.31 | 82.9 | 13 Dec | 0.09 | 98.5 |
| 14 Mar | 0.06 | 5.6 | 14 Jun | 0.53 | 30.1 | 14 Sep | 0.31 | 83.2 | 14 Dec | 0.09 | 98.6 |
| 15 Mar | 0.06 | 5.6 | 15 Jun | 0.53 | 30.6 | 15 Sep | 0.30 | 83.5 | 15 Dec | 0.09 | 98.6 |
| 16 Mar | 0.06 | 5.7 | 16 Jun | 0.53 | 31.2 | 16 Sep | 0.29 | 83.8 | 16 Dec | 0.09 | 98.7 |
| 17 Mar | 0.07 | 5.8 | 17 Jun | 0.53 | 31.7 | 17 Sep | 0.29 | 84.1 | 17 Dec | 0.09 | 98.8 |
| 18 Mar | 0.07 | 5.8 | 18 Jun | 0.52 | 32.2 | 18 Sep | 0.28 | 84.4 | 18 Dec | 0.08 | 98.9 |
| 19 Mar | 0.07 | 5.9 | 19 Jun | 0.52 | 32.7 | 19 Sep | 0.28 | 84.6 | 19 Dec | 0.08 | 99.0 |
| 20 Mar | 0.08 | 6.0 | 20 Jun | 0.52 | 33.3 | 20 Sep | 0.27 | 84.9 | 20 Dec | 0.08 | 99.1 |
| 21 Mar | 0.08 | 6.1 | 21 Jun | 0.53 | 33.8 | 21 Sep | 0.27 | 85.2 | 21 Dec | 0.08 | 99.1 |
| 22 Mar | 0.08 | 6.1 | 22 Jun | 0.53 | 34.3 | 22 Sep | 0.27 | 85.4 | 22 Dec | 0.08 | 99.2 |
| 23 Mar | 0.08 | 6.2 | 23 Jun | 0.53 | 34.9 | 23 Sep | 0.26 | 85.7 | 23 Dec | 0.08 | 99.3 |
| 24 Mar | 0.08 | 6.3 | 24 Jun | 0.54 | 35.4 | 24 Sep | 0.26 | 86.0 | 24 Dec | 0.08 | 99.4 |
| 25 Mar | 0.08 | 6.4 | 25 Jun | 0.54 | 35.9 | 25 Sep | 0.26 | 86.2 | 25 Dec | 0.08 | 99.5 |
| 26 Mar | 0.08 | 6.5 | 26 Jun | 0.55 | 36.5 | 26 Sep | 0.26 | 86.5 | 26 Dec | 0.08 | 99.6 |
| 27 Mar | 0.08 | 6.6 | 27 Jun | 0.55 | 37.0 | 27 Sep | 0.26 | 86.8 | 27 Dec | 0.09 | 99.6 |
| 28 Mar | 0.08 | 6.6 | 28 Jun | 0.56 | 37.6 | 28 Sep | 0.26 | 87.0 | 28 Dec | 0.09 | 99.7 |
| 29 Mar | 0.08 | 6.7 | 29 Jun | 0.57 | 38.2 | 29 Sep | 0.26 | 87.3 | 29 Dec | 0.09 | 99.8 |
| 30 Mar | 0.08 | 6.8 | 30 Jun | 0.57 | 38.7 | 30 Sep | 0.26 | 87.5 | 30 Dec | 0.09 | 99.9 |
| 31 Mar | 0.07 | 6.9 | | | | | | | 31 Dec | 0.09 | 100.0 |