# Peer review of "Rain erosivity map for Germany derived from contiguous radar rain data"

_Hydrology and Earth System Sciences, 2018_

## Referee Comment (RC1) · A. Vrieling (Referee) · 24 Oct 2018

I really like the fact that the authors have used rainfall radar data to map rainfall erosivity at the national scale. This is a great follow up from earlier papers that advocated the use of gridded rainfall estimates for this purpose, with the present ground-based radar data having a clear advantage over satellite-derived data in terms of their accuracy and spatial and temporal resolutions. As such, I much support its publication. Nonetheless, I do have a number of concerns regarding the methodology and also the write-up of the work, which at times is unclear and poorly structured. The main concerns are:

1. My main issue with the manuscript is its poor readability. Many statements are unclear, often lacking precision; for example a reader sometimes needs to guess what

data the authors refer to precisely. Further, I also note a mixing of results in the methods section, and a repetition of methods in the discussion section. I realize that my statement regarding the poor readability is rather general, but I try to specify as good as I can specific instances in the detailed comments section below. However, in general the feeling I obtained was that the authors should do one step back from their research and make their text more accessible by taking more a perspective of readers that are less familiar with what the authors did. Finally, also the alignment between section headings in methods and results could be better for easier understanding of what results belong to which methods.

2. The main analyses that result in the erosivity map were performed using 1-hr radar data. This is partially justified by the authors due to the amount of data to be processed (P5L9-12). While data reduction can be an advantage for calculations, I still wonder though whether this is the best effort possible. Rainfall erosivity is by definition dictated by intensity and intensity is much better captured with 5-minute data. The reported advantage of the adjustment to rain gauge measurements could also hold for 5-minute data, i.e. it should be easy to re-assign the 1-hr adjusted data to the 5-minute intervals. This leaves me wondering if we would not get to better estimates if we take advantage of the 5-minute intervals. I agree that data storage and processing requirements will increase, but with a smart computer code it should not be too hard to calculate through 17 years of 5-minute data. While I do not necessarily request the authors to change this now (although I would applaud it), I would at least expect a discussion as to whether future improvements of their map is possible, given also that they admit that "high intensity peaks" (P13L4) are very important for erosivity.

3. I do not fully understand why the authors chose to present erosivity at the daily time scale, and this raises two questions. A) Yes, we know that erosivity is stochastic so the scatter in Figure 5 is hardly a surprise. However, the main question eventually is how erosivity is combined with other factors (e.g. vegetation) to estimate erosivity. Arguably this could be at the daily time scale, but also weekly or monthly could provide sufficient

temporal detail and as an additional advantage would have a smoothing effect in itself. B) Because the paper partially focuses on daily erosivity, I was curious to know how the authors dealt with night-time rain events. A storm can occur overnight thus belonging to two days. Nonetheless, as a single event, this should be treated accordingly as such when aggregating for a year. Any insight in this would help.

4. Although I understand part of the reasons for smoothing, particularly for presenting average annual erosivity, I found little justification for the methods used in this study. Rather mechanistically it is described what is done, but reasons are mostly lacking. An example of this is P7L9-15: a sequence of three temporal filters are applied. I wonder whether a single carefully chosen filter would not suffice. The poor justification is also true for the procedure described in P5L20-22 on scaling: how were those parameters determined?

5. I wonder why the authors' main interest seems to obtain a smooth average annual erosivity map. This leaves me wondering what they see as the main application of this map; the general statement in P1L25 (and P13L20) makes some sense but could be further elaborated. I would argue that if the interest is mostly (or partially also) in erosion monitoring, we may not need any smoothing at all, but rather we want to know when, where, and to what extent a surface is exposed to erosive rainfall. In this case we would not want to smooth out the stochastic nature of the erosivity, but rather retain it, because it offers important insights on erosion occurrence and could directly be combined with temporal vegetation assessments (e.g. from satellites). While I do not mean to necessarily promote own work, the discussions in https://doi.org/10.1016/j.gloplacha.2014.01.009 could be of interest in this regard (in which actually I also stated the potential interest of ground-based weather radars for the purpose of erosivity assessment!).

6. Despite some of the comments above, Figure 5 is an important result in my view. I understand that this is an average within-year distribution for Germany, but I miss a discussion (and/or results) that show the seasonal distribution for sub-regions. Perhaps

it would also be an idea to show monthly maps?

7. The authors frequently refer to the R map of Sauerborn (1994). It would be helpful if this map could be provided (e.g. in the Appendix) using the same color scheme as used for the other maps.

8. A trend in erosivity is proving through comparison with the older map, and (luckily) also comparing within existing rainfall stations. Because the authors also have a 17-yr series of erosivity, I wondered if that could also be a basis to say something about a (spatially-aggregated) time series for that period. Particularly as the authors report a large trend in the last few decades (P12L5).

9. Station data are used in this study, but not to provide a direct validation of erosivity measures it seems. I would think that this could be a nice addition, i.e. to evaluate possible discrepancies between rain radar derived erosivity and station-derived erosivity.

Detailed comments: - P1L12: "for the first time". This seems incorrect as the authors also published work before that uses rain radar data to assess erosivity.

- P1L14: "extraordinarily large filtering"; this is a vague statement that needs rewording

- P1L15: "averaging 2001 to 2017" is not a precise statement. Probably the authors mean that the annual erosivity of 2001 to 2017 was averaged?

- P1L16 (and also L19/20): "the previous map" should be rephrased: "the erosivity map currently used in Germany, which is based on . . ."

- P1L20: unclear: do the authors mean to say that this is based on stations that were available in 1960-1980 and continue to report until present?

- P1L21: "by weather changes that may already be . . . 1970s." Avoid emotional wordings like "dramatic"; rather state that "but by a change in climatic conditions".

- P1L22: erosivity does not "fall"

- P1L22-25: I have the feeling that this issue is a bit overstated: still the erosivity during winter months is rather small. I suppose that this requires a joint analysis with vegetation cover, which is outside the scope of this paper. Probably the main erosion on cropland would still occur during spring (in May farmers in NW-Germany have only just planted maize for example) and late summer when crops like wheat are harvested. Now it is not possible to state that "for many crops" we have "higher erosion" (P1L22)

- P1L25: the "thus" and "definite" suggest a very strong causal relationship between previous statements and what is said here. I think that the authors should be more cautious; while an important input, the work cannot make definite conclusions about erosion yet.

- P2L4-6: this is a bit vague; applied and used by whom? Does this refer to Germany or more general?

- P2L29-30: I would at least shortly acknowledge existing efforts to do the same with gridded data of satellite-derived rainfall estimates.

- P3L10-16: I feel somewhat uncomfortable with the present tense of "expect" here, given that this article is a report of a work already completed. I suggest removal, but highlighting this in the results/discussion.

- P3L19-26: I wonder if there may be any effect of changes in the network/systems on the erosivity estimates?

- P4L8: RW is an acronym for what?

- P4L25: other versions of this equation exist. See also: "van Dijk AIJM, LA Bruijnzeel, & CJ Rosewell (2002). Rainfall intensity-kinetic energy relationships: A critical literature appraisal. Journal of Hydrology 261: 1-23". Why did the authors choose for this equation?

- P5L23: section 2.2 should be 2.3 (and also for next sections numbering should continue accordingly)

- P6L7/11: "neighbor" should read "neighboring"

- P6L8: "krige" should read "kriging"

- P6L11: "This . . . pattern". Sentence unclear.

- P6L31-32: this seems to be out of place, as it reports on results.

- P7L12: "and the level shifts in the smooth"? I do not understand the sentence

- P7L13: Hanning with capital H?

- P7L14-15: I think that after two reads I get the meaning, but could be formulated clearer.

- P7L16-19: this seems to belong to results also.

- P7L26: if so, I would expect a clearer proof of this, e.g. by linking height from a DEM to erosivity, or at least show a DEM of Germany somewhere in the paper.

- P8, Section 3.2. I fail to clearly see a main message appearing from this section; what is the key lesson/result that the authors want to convey?

- P8L7: strange combination of present and past tense (smoothed). Specify that the 10-15km refer to this study.

- P8L7-9: the "disappearance" is quite logical from the description of winsorizing.

- P8L11: "Rain erosivity from 5-min resolution data ..": it is unclear what erosivity this refers to: annual? Is this for 2011?

- P8L15-16: probably this is described in methods but a bit unclear why 2011 and 2012 were chosen here.

- P9L32: "extreme" and "violent" sounds rather exaggerated. A more scientific formulation would be appreciated.

- P10L17-18: the previous sentences compared radar-based erosivity with meteostation erosivity. Therefore I believe that this sentence makes little sense here, because there may in fact be differences because of the different data used. I more trust the fact that in P10L19 the same stations (and hopefully the same methods) were used. So please do not conclude when the previous statements do not support the conclusion yet.

- P10L29-30: location is not so clear: could it be indicated in Figure 1?

- P11L3-4: perhaps it is me, but I fail to fully grasp what regression was precisely done here (what against what), and with what purpose?

- P11L9: "trains" should be "rains"?

- P11L21: see previous points also: I think that this "most pronounced changes" is overstated: the erosivity is still small.

- P12L28-P13L2: this seems too much repetition of results to me.

- P13L2: link between sentence ending with "resolutions" and sentence starting with "The pronounced" is not clear; this seems to be another topic.

- P13L3-4 I do not understand this: how can orographic rainfall increase hourly but not the peaks? And with peaks the authors refer to sub-hourly?

- P13L16: expected by whom? Rather "than what existing erosivity maps showed"

- Captions should be self-explanatory: in Figure 1 I do not understand the "for a range of 128km AND the 2017 configuration". Please revise to make the caption clearer.

- Figure 2 caption: "Erosivity map" specify in caption if this is annual average erosivity. Also units should be reported in caption.

- Figure 3: also here it should be clarified if we are looking at annual erosivity, and which years of data and why. Also why do we see a lag up to 50km? Would it make sense to make it longer? Why or why not?

- Figure 5: caption should specify if daily erosion index (circles and blue) is from radar data.

- Figure 6: caption should specify if Sauerborn used the exact same methods

- Table 1: last two entries (1h winsorized and kriged) are also 17-yr? Specify. Also what years are used for annual/biannual?

- Table A2 is probably not needed because Figure 5 is presented. If still kept, it should be organized differently so that Jan-Apr are on one page.

- Take care with wording of high/higher/low/lower: usually this refers to altitude, whereas other parameters/values are small or large.

---

## Referee Comment (RC2) · Anonymous Referee #2 · 29 Oct 2018

This review is for the article "Rain erosivity map for Germany derived from contiguous radar rain data" by Fischer et al. In this work the authors use 17 years of gauge-corrected radar data to produce values of erosivity at 1 km$^2$ resolution over Germany. The data were noisy so significant data treatment was applied to produce a "typical" erosivity map and an annual cycle of erosivity. The new values show greater erosivity than previously produced maps and the seasonal distribution shows an increase in winter erosivity. The main advantage of the new approach is the use of continuous data over the region.

**General comments**

The article is very well written, the analyses are clearly described and figures are well chosen. The results will clearly be of use.

However, some of the choices made in data treatment require further justification. My primary concern is about the level of data treatment that has been applied, which is, as the authors state, "extraordinarily large". Because the amount of smoothing applied is indeed more than normal, it should be carefully justified.

The aim is to produce a map of "typical" erosivity over Germany, but the erosivity distributions in time are skewed and contain outliers (from rare, extreme events) that make finding one representative value per pixel a challenge. A related problem is possible sampling effects, meaning differences between the sampled and true distribution of values (the authors mention this with respect to measurements from gauge networks that may miss entire events). The authors have applied data transformation techniques to find typical values, smooth them in space, and smooth the evolution of the average erosion index over time.

I'd like to comment on each data transformation undertaken, first to produce the per-pixel values:

1. **Winsorizing:** For each pixel, the mean erosivity over 17 yearly values is taken using winsorizing. In this case the authors only replace the lowest and the highest value (with the second-lowest and second-highest respectively). How was the choice made to use winsorizing over, say, the sample median? The choice of the method used (e.g. sample median or winsorizing, and the amount of winsorizing used) should be justified – for example through the use of a density plot of erosivity values, in which the skewness will be clearly visible, to show that the final values produced are representative of "typical" values of erosivity.

2. **Bias-correction:** The authors state that the winsorized mean is biased for long-tailed variables. But, for skewed distributions the winsorized mean should be closer to a "typical value" of the population than the sample mean because of the removal of outlier values. So is it not the case that winsorizing produces a less biased estimate of a central tendancy than the sample mean, and the bias correction suggested by the authors undoes the benefit of the winsorizing by matching back to the (spatial) sample means which are themselves affected by outliers?

3. **Ordinary kriging:** Kriging is used to fill gaps not covered by the radar data (due to beam-blocking, for example), and block kriging is used to smooth the output field. Kriging requires at least roughly symmetrically distributed input data (ideally they would be normally distributed) so that mean values are representative. It should therefore be mentioned in the article whether the distribution of "typical" values after the winsorizing procedure is symmetric, and if not whether steps have been taken to correct for this (possible options are a log transformation and/or use of the Cressie variogram estimator). Block kriging is being used in a non-standard way, as a smoother, so that each $1 \times 1$ km$^2$ pixel is estimated as the mean of values across a $10 \times 10$ km$^2$ block. How was the block size of $10 \times 10$ km$^2$ chosen?

After the spatial processing, the annual cycle of erosivity is calculated. Afterwards, smoothing was applied to the daily timeseries of averages. Again commenting on each step:

1. **Daily erosion index:** The erosion index is calculated for each pixel and then averaged across space for each day of the year. It was not clear to me whether the pixel values used to make this average were treated in any way (kriged perhaps?) or were raw 1 km$^2$ values (I assume it was the raw values so that they

were daily). If the distribution of daily EI values (across space for each day) is heavily skewed, then the mean of their values may not be representative (the median or a winsorized mean, for example, may be better). Was any testing for this done?

2. **13-day centred median, 3-day skip mean, and 25-day centred hanning mean:** This choice of smoothing routines needs to be better justified. Why was this combination of window sizes (13, 3, and 25 day) and operators chosen, and how was it judged whether the smoothed values represented the true signal?

As a suggestion that may provide more information to the reader: the authors could consider displaying maps not only of winsorized mean of annual values, but also per-pixel median, 10th percentile, and 90th percentile, to show not only "typical" values but maps of extreme erosivity values as well, and to show the spread of values for each pixel.

**Specific comments**

1. Page 2, line 24–25: "Unstable and unreliable transfer functions result that differ pronouncedly" – I do not understand the sentence, could you please rephrase?

2. Page 2, line 30: Please include a general reference for the radar measurement principle. One such reference could be the book by Bringi and Chandrasekar (2001), *Polarimetric Doppler Weather Radar*, Cambridge Uni. Press.

3. Page 3, line 12: I see your point that the use of continuous radar data avoids missing large and rare events that could be missed completely by gauge networks. But you do a lot of processing to the radar data, including winsorizing and smoothing, which reduces the influence of rare extreme events on the summary

statistics. There are two separate problems here: sampling (gauges may miss an event) and then what value to use as a "typical" value for a skewed distribution. It is important that justifications for the data treatment show that the chosen "typical" measure is appropriate.

4. Page 3, line 31: Which $Z$-$R$ relationship is used?

5. Page 4, line 6: Is the figure of 1 gauge per 80 km$^2$ an average value?

6. Page 4, line 20: For clarity, it would be helpful to include the units of $I_{\mathrm{max30}}$ and $E_{\mathrm{kin}}$ when the variables are introduced here; this is especially important because $E_{\mathrm{kin}}$ [kJ m$^{-2}$] and $E_{\mathrm{kin},i}$ [kJ m$^{-2}$ mm$^{-1}$] have different units.

7. Page 4, line 24: You should reference Fischer et al 2018 (from your references list) here since your definitions, units, and descriptions are very similar to those used in your previous paper.

8. Page 4, line 29: "the $R_e$ sum" – do you mean "the sum of $R_e$"?

9. Page 5, lines 14–15: For a given pixel, if too many years were excluded then the sampling may become less representative. How often were pixels affected by this exclusion of years, and were there pixels for which many years were excluded?

10. Page 5, lines 16–22: "replaced by the maximum 1-h rain depth" - should this read rain intensity?

11. Page 5, lines 16–22: As I understand it, the scaling factors are being used to adjust the method of calculating erosivity to put a "virtual rain gauge" in each radar pixel, to account for the fact that radar measurements are areal and integrated over time and therefore smooth out rainfall intensity peaks. Since rain intensity depends on temporal resolution, and you require 30 minute maximum rain rates

which would be smoothed by the use of 1 hour radar data, I see why a temporal scaling factor could be used. But spatially, the areal measurements at 1 km$^2$ resolution can be assumed to be representative of each 1 km$^2$ pixel, and since you are producing erosivity values at the same resolution, I don't understand why a spatial scaling factor (or indeed the method scaling factor) is required. Please could you explain more here why the scaling factors are used and how they are applied (e.g. it is not clear which threshold is lowered to 5.8 mm h$^{-1}$).

12. Page 6, line 14: The use of some independent data to test the spatial representativity of the smoothed data is a good idea, but is this test data independent? It is also based on radar data. Has the test region data been compared to gauges or other ground truth data to ensure it is accurate?

13. Page 7, lines 1–2: "The cumulative distribution curve for the test region calculated from 5-min data will then be a fair estimate of the return periods anywhere in the research area" – I do not think this is proven. Even if the test region and the whole area agree at 1 hour resolution, extreme intensities are smoothed out at this lower resolution, so it does not necessarily follow that the 5 minute cumulative distributions are the same across all regions.

14. Page 7, lines 19–20: Please include a reference for these statements about radar accuracy (they are correct but require a citation).

15. Section 3.1: I suggest that to back up your observation that the regional pattern in erosivity is dominated by orography, you should include a topographic map showing ground elevation for comparison with the map in Figure 2.

16. Page 8, line 13: "Using the normal distribution" – but are the erosivity values normally distributed?

17. Page 8, line 32: Which variogram was the kriging conditioned by? I would expect

kriging to maintain the spatial stucture even in a block kriging case, so it is odd that the kriging changes the variogram.

18. Section 3.3: I suggest adding more lines to your Figure 4 to show all the lines you mention in the text.

19. Page 9, line 21: I'm surprised that you would expect less than the mean erosivity for an event with a return period of 2 years. Any comment there?

20. Page 9, line 27: "d" is presumably for days but should be spelled out.

21. Page 9, line 32: To see exactly what is going on here, did you compare the the distributions of erosivity values for each of these example days? I suspect that the median values would be more stable.

22. Page 10, line 20: I think Fig. 5 should be Fig. 6.

23. Page 11, line 18: No definition of C-factor calculations is given; please add one.

24. Page 12, line 22: Please define (R)USLE.

25. Figures 2, A1, and A2: Units for the plotted variable should be stated either in the key or caption.

**Technical corrections**

- Page 5, line 19: The word "occurred" can be removed.

- Page 7, line 12: By "in the smooth" do you mean "in the smoothing operation"?

- Figure 2: In the caption the average sizes of the local authority and community areas should be areas ($km^2$).

[Figure]

- Page 8, line 8: "very extreme" is redundant when "extreme" will do.

- Page 9, line 23: "extremer" should be replaced by "more extreme".

---

## Author Comment (AC1) · 10 Dec 2018

A. Vrieling (Referee)

a.vrieling@utwente.nl

I really like the fact that the authors have used rainfall radar data to map rainfall erosivity at the national scale. This is a great follow up from earlier papers that advocated the use of gridded rainfall estimates for this purpose, with the present ground-based radar data having a clear advantage over satellite-derived data in terms of their accuracy and spatial and temporal resolutions. As such, I much support its publication. Nonetheless, I do have a number of concerns regarding the methodology and also the write-up of the work, which at times is unclear and poorly structured. The main concerns are:

1. My main issue with the manuscript is its poor readability. Many statements are un- clear, often lacking precision; for example a reader sometimes needs to guess what data the authors refer to precisely. Further, I also note a mixing of results in the methods section, and a repetition of methods in the discussion section. I realize that my statement regarding the poor readability is rather general, but I try to specify as good as I can specific instances in the detailed comments section below. However, in general the feeling I obtained was that the authors should do one step back from their research and make their text more accessible by taking more a perspective of readers that are less familiar with what the authors did. Finally, also the alignment between section headings in methods and results could be better for easier understanding of what results belong to which methods.

We carefully edited the manuscript in order to improve its readability and to remove any ambiguity.

2. The main analyses that result in the erosivity map were performed using 1-hr radar data. This is partially justified by the authors due to the amount of data to be processed (P5L9-12). While data reduction can be an advantage for calculations, I still wonder though whether this is the best effort possible. Rainfall erosivity is by definition dictated by intensity and intensity is much better captured with 5-minute data. The reported advantage of the adjustment to rain gauge measurements could also hold for 5-minute data, i.e. it should be easy to re-assign the 1-hr adjusted data to the 5-minute intervals. This leaves me wondering if we would not get to better estimates if we take advantage of the 5-minute intervals. I agree that data storage and processing requirements will increase, but with a smart computer code it should not be too hard to calculate through 17 years of 5-minute data. While I do not necessarily request the

authors to change this now (although I would applaud it), I would at least expect a discussion as to whether future improvements of their map is possible, given also that they admit that "high intensity peaks" (P13L4) are very important for erosivity.

We fully agree that we would get better results for maps of PAST erosivity (= "hindcast" erosion modelling) if we would use 5-min data and we have shown this in Fischer et al. (2018). Usually such maps are useful for shorter time scales (event, year) for comparison with recorded erosion damages. Long-term average maps are usually applied for PLANNING (= "forecast" erosion modelling). In this case we do not need the exact location of a thunderstorm cell in the past but the general pattern that can be expected. This requires smoothing of the stochastic locations.

We added an extensive overview over the two types of R factor use and the smoothing that is inherent in present R maps.

3. I do not fully understand why the authors chose to present erosivity at the daily time scale, and this raises two questions.

This is a misunderstanding. We did not consider days (see below, your Point 3.B). Days were only considered for the seasonal EI distribution where it is necessary. In this case an event, even if it extended over several days, was assigned to that calendar date that was in the middle between the start and the end of the event.

A) Yes, we know that erosivity is stochastic so the scatter in Figure 5 is hardly a surprise. However, the main question eventually is how erosivity is combined with other factors (e.g. vegetation) to estimate erosivity. Arguably this could be at the daily time scale, but also weekly or monthly could provide sufficient temporal detail and as an additional advantage would have a smoothing effect in itself.

Yes, this is the basis for the crop stage period of the C factor. However, the crop stage periods between crops appear at different dates and current developments (RUSLE) allow for a continuous calculation of the soil loss ratio. A classification of the seasonal distribution (e.g. monthly means) will always decrease the accuracy of C factor calculations (except for the unlikely case that crop stages change only at the beginning of months) compared to daily values (note that these daily values are not daily events but the fraction of annual erosivity that on average can be expected to occur at a certain day).

B) Because the paper partially focuses on daily erosivity, I was curious to know how the authors dealt with night-time rain events. A storm can occur overnight thus belonging to two days. Nonetheless, as a single event, this should be treated accordingly as such when aggregating for a year. Any insight in this would help.

We did not distinguish between day-time and night-time but strictly followed the recommendations by Wischmeier. Often the events extended over midnight (because they often start in late afternoon and end in the early morning hours) or they extended even over

several days. Our calculations did not use days but continuous temporal records over 17 years. We clarified this in the text.

4. Although I understand part of the reasons for smoothing, particularly for presenting average annual erosivity, I found little justification for the methods used in this study. Rather mechanistically it is described what is done, but reasons are mostly lacking. An example of this is P7L9-15: a sequence of three temporal filters are applied. I wonder whether a single carefully chosen filter would not suffice. The poor justification is also true for the procedure described in P5L20-22 on scaling: how were those parameters determined?

We added to the Introduction a better description of the twofold and contrasting applications of R maps and a description of the strong smoothing that has been unintentionally applied in previous maps. Furthermore, we replaced 'filtering' by 'smoothing' because filtering is often understood as removing certain (regular) frequencies while the occurrence of erosion events is stochastic.

To answer the question more specifically: An individual smoothing algorithm did not work (and cannot work). Even if one would exist, there would be no advantage for the reader or user of the data. The length of description of the method overemphasizes the importance of smoothing. The cumulative distribution function of the raw data correlates with the cumulative distribution function of the smoothed data with $r^2=0.9998$ (n=365). This information was added to the manuscript

5. I wonder why the authors' main interest seems to obtain a smooth average annual erosivity map. This leaves me wondering what they see as the main application of this map; the general statement in P1L25 (and P13L20) makes some sense but could be further elaborated. I would argue that if the interest is mostly (or partially also) in erosion monitoring, we may not need any smoothing at all, but rather we want to know when, where, and to what extent a surface is exposed to erosive rainfall. In this case we would not want to smooth out the stochastic nature of the erosivity, but rather retain it, because it offers important insights on erosion occurrence and could directly be combined with temporal vegetation assessments (e.g. from satellites). While I do not mean to necessarily promote own work, the discussions in https://doi.org/10.1016/j.gloplacha.2014.01.009 could be of interest in this regard (in which actually I also stated the potential interest of ground-based weather radars for the purpose of erosivity assessment!).

We added to the Introduction a better description of the twofold and contrasting applications of R maps and a description of the strong smoothing that has been unintentionally applied in previous maps.

6. Despite some of the comments above, Figure 5 is an important result in my view. I understand that this is an average within-year distribution for Germany, but I miss a discussion (and/or results) that show the seasonal distribution for sub-regions. Perhaps it would also be an idea to show monthly maps?

We added a sentence and a figure in the appendix to show the (non-existent) regional variation.

"There was no detectable difference in the seasonal variation between different regions in Germany (see Fig. A5 in the Appendix). The cumulative density functions of different regions correlated with at least $r^2 = 0.998$ (n = 365)."

7. The authors frequently refer to the R map of Sauerborn (1994). It would be helpful if this map could be provided (e.g. in the Appendix) using the same color scheme as used for the other maps.

We added it as Fig. A2

8. A trend in erosivity is proving through comparison with the older map, and (luckily) also comparing within existing rainfall stations. Because the authors also have a 17-yr series of erosivity, I wondered if that could also be a basis to say something about a (spatially-aggregated) time series for that period. Particularly as the authors report a large trend in the last few decades (P12L5).

We added to the discussion

A time series of 17 yr is regarded too short in meteorology for calculating temporal trends. The data in Sauerborn (1994) were derived from different periods for different states. If we calculate the state-wide mean R factors from her transfer functions relative to the state-wide mean R factors of the radar-derived map and plot this relative R factor against the mean year from which the state-specific data originated, a 23-yr long period can be covered by the means (Fig. 7; years < 1990; the total time period of individual years covers an even wider range, mostly about ± 5 yr around the mean year). During this period there was a slight but insignificant increase in erosivity with time. This increase smoothly leads over to the steeper increase in relative R for the radar-derived Germany-wide annual R factors if we express them again relative to the 17-yr mean (Fig. 7; years > 2000). Combining both data sets covers more than 60 years and yields a very highly significant regression ($r^2 = 0.6388$, n = 28). This regression indicates that at the end of the radar time series (2017) the R factor likely is already 20% higher than the values depicted in Fig. 2. There was no offset between both time series, which could indicate that high values obtained from the radar data are caused by the differing method. Rather, this time series again corroborates that the large increase in R is not a methodological artefact but due to accelerating climate change.

[Figure]

Fig. 7: Average R factor relative to the mean radar-derived R factor depending on the mean year of data origin. Data below year 1990 are calculated from state-wide averages determined from meteorological station records; year is the mean year of station records. Data above year 2000 are radar-derived R factors of entire Germany for individual years. The closed circle denotes the reference point (present map).

9. Station data are used in this study, but not to provide a direct validation of erosivity measures it seems. I would think that this could be a nice addition, i.e. to evaluate possible discrepancies between rain radar derived erosivity and station-derived erosivity.

Fig. 6 provides a direct comparison. A more detailed analysis (e.g., regional differences) seems not appropriate because the number of 33 stations and the unavoidable scatter in the data do not allow splitting the data. We also do not expect a large regional variation given the similarity between the new map and the old map (which we now provide in the Appendix). Finally, a comparison between station derived erosivities and radar derived erosivities has already been published for a large data set (115 stations and 19 944 events) by Fischer et al. (2018). In this publication, in which we neither intended to create a map nor to compare with the old map, we were not restricted to the 'Sauerborn' stations and thus could use 115 stations.

Detailed comments:

- P1L12: "for the first time". This seems incorrect as the authors also published work before that uses rain radar data to assess erosivity.

Rephrased

- P1L14: "extraordinarily large filtering"; this is a vague statement that needs rewording

We rephrased this in the Introduction and we added to the Discussion:

This pronounced stochasticity is due to the small size of convective rain cells. Just recently it has been shown by analysing radar derived rain pattern of the largest rainfall events that on average the rain amount is halved within a distance of only 2 km around the central point of a rain cell (Lochbihler et al., 2017). Given that rain amount is squared in the calculation of rain

erosivity, the R factor decreases to one fourth within this distance. Larger areas can only be covered if there is movement of the rain cells. This small size of rain cells questions the use of rain gauges that only sparsely can cover space to derive rain erosivity. The inconsistent transfer functions among German states to derive erosivity from rainfall maps likely originated in the high stochasticity of rain gauge measurements under such conditions. It was only the unintended but unavoidable smoothing that was inherent in previous approaches that allowed deriving such maps. Radar technology enables us to replace this unintended smoothing by clearly defined statistical protocols and to quantify the effect of smoothing.

Lochbihler, K., Lenderink, G., Siebesma A.P.: The spatial extent of rainfall events and its relation to precipitation scaling, Geophys. Res. Lett., 44, 8629–8636 (2017). doi:10.1002/2017GL074857

- P1L15: "averaging 2001 to 2017" is not a precise statement. Probably the authors mean that the annual erosivity of 2001 to 2017 was averaged?

Changed as suggested

- P1L16 (and also L19/20): "the previous map" should be rephrased: "the erosivity map currently used in Germany, which is based on ... "

Changed as suggested; we introduced the name "Sauerborn map" in this place to replace "the erosivity map currently used in Germany" in the following occurrences.

- P1L20: unclear: do the authors mean to say that this is based on stations that were available in 1960-1980 and continue to report until present?

We rephrased the sentence to make it clearer: "This increase in erosivity was confirmed by long-term data from rain gauge stations that were used for the previous map and which are still in operation."

- P1L21: "by weather changes that may already be ... 1970s." Avoid emotional wordings like "dramatic"; rather state that "but by a change in climatic conditions".

Reworded

- P1L22: erosivity does not "fall"

Reworded

- P1L22-25: I have the feeling that this issue is a bit overstated: still the erosivity during winter months is rather small. I suppose that this requires a joint analysis with vegetation cover, which is outside the scope of this paper. Probably the main erosion on cropland would still occur during spring (in May farmers in NW-Germany have only just

planted maize for example) and late summer when crops like wheat are harvested. Now it is not possible to state that "for many crops" we have "higher erosion" (P1L22) - P1L25: the "thus" and "definite" suggest a very strong causal relationship between previous statements and what is said here. I think that the authors should be more cautious; while an important input, the work cannot make definite conclusions about erosion yet.

We show (although we do this only in the discussion) that erosion under winter wheat (which is the most often grown crop in Germany) will be four-fold higher than previously expected. Practically all other crops (except clover-grass and except the system of "mulch tillage") also pass winter in a susceptible state (which is well known; we give the reference). We do not see any exaggeration or speculation in these statements.

- P2L4-6: this is a bit vague; applied and used by whom? Does this refer to Germany or more general?

We have expanded the description of how R factor maps were usually derived in the past

- P2L29-30: I would at least shortly acknowledge existing efforts to do the same with gridded data of satellite-derived rainfall estimates.

We included two references to this method (Vrieling et al., 2010, 2014)

- P3L10-16: I feel somewhat uncomfortable with the present tense of "expect" here, given that this article is a report of a work already completed. I suggest removal, but highlighting this in the results/discussion.

These are our hypotheses (and two out of three turned out to be wrong). We replaced "expect" and the present tense by "Our hypothesis was".

- P3L19-26: I wonder if there may be any effect of changes in the network/systems on the erosivity estimates?

We have used the RADKLIM data set that is the best estimate of precipitation based on radar- and gauge-data in Germany. A sophisticated quality control, the merging of different data sources, and a reprocessing applying one software tool lead to an at most homogenous data set, where the influence of network changes on precipitation estimates is eliminated or at least strongly reduced. However, a distinct improve in quality is detectable due to improved quality control of the raw data.

The changes in the measuring system over time were mainly intended to improve measurements where former locations had specific deficits that became apparent over time. We insofar expect the later measurement to be better, also because of the technical developments in radar technology. Furthermore, we expect that the change of the locations improves long-term averages compared to long-term averages with fixed locations because local deficits level out. These improvements have been documented for precipitation (mainly

in internal reports). How much they improve erosivity calculation is unknown. An evaluation for erosivity would be rather tricky (because of the much larger variability of erosivity compared to precipitation) and of little general interest because this would mainly reflect local effects and it would only be applicable to the past. Restricting our data to the latest configuration would also not be an improvement because of the shorter time series and because the restrictions given by the latest   configuration would then not be dampened by measurements with the older configuration.

We did not expand on this in the text because (i) there is too much speculation, (ii) this is a different topic that would distract from our topic

- P4L8: RW is an acronym for what?

RW is an abbreviation of **R**ADOLAN respective **R**ADKLIM and **W**eighted as it is a weighted sum of two products adjusted by different methods. We use RW only as a name but not as an acronym to indicate, which radar data were used. The data set is freely available and can be found by this name. Otherwise a lengthy explanation would become necessary. This explanation can be found in Winterrath et al. (2018), whom we cite.

Winterrath, T., Brendel, C., Hafer, M., Junghänel, T., Klameth, A., Lengfeld, K., Walawender, E., Weigl, E., Becker, A.: RADKLIM Version 2017.002: Reprocessed gauge-adjusted radar data, one-hour precipitation sums (RW) DOI: 10.5676/DWD/RADKLIM_RW_V2017.002, 2018.

- P4L25: other versions of this equation exist. See also: "van Dijk AIJM, LA Bruijnzeel, & CJ Rosewell (2002).  Rainfall intensity-kinetic energy relationships:  A critical literature appraisal.  Journal of Hydrology 261:  1-23".  Why did the authors choose for this equation?

We added a justification:

"We used the equation by Wischmeier and Smith (1978) to calculate kinetic energy although several others have also been proposed (van Dijk et al., 2002) with none being superior (Wilken et al. 2018). Our choice retained comparability with the Sauerborn map. Furthermore van Dijk et al. (2002) had shown that kinetic energy as obtained by the Wischmeier-and-Smith equation did not deviate from measured kinetic energy in Belgium neighboring Germany."

Another reason (not mentioned in the manuscript) is that the equation by Wischmeier and Smith (1978) is defined as standard among German authorities. Only recently this has been re-affirmed (DIN, 2017). An R map that is not based on the defined standard would not be used by German authorities.

DIN – Deutsches Institut für Normung: DIN 19708: 2005-02 Bodenbeschaffenheit – Ermittlung der Erosionsgefährdung von Böden durch Wasser mit Hilfe der ABAG. Beuth-Verlag, Berlin, 2017.

- P5L23: section 2.2 should be 2.3 (and also for next sections numbering should continue accordingly)

Error corrected

- P6L7/11: "neighbor" should read "neighboring"

Replaced

- P6L8: "krige" should read "kriging"

Replaced

- P6L11: "This ... pattern". Sentence unclear.

We have expanded the sentence to better convey its message

- P6L31-32: this seems to be out of place, as it reports on results.

We added a reference to indicate that this is *a priori* knowledge (although the same was later found in our results)

- P7L12: "and the level shifts in the smooth"? I do not understand the sentence

This wording had been taken from the cited statistical reference. Now we reworded the sentence and used common language.

- P7L13: Hanning with capital H?

Changed as suggested

- P7L14-15: I think that after two reads I get the meaning, but could be formulated clearer.

Reworded and expanded

- P7L16-19: this seems to belong to results also.

We reworded this paragraph in order not to anticipate results.

- P7L26: if so, I would expect a clearer proof of this, e.g. by linking height from a DEM to erosivity, or at least show a DEM of Germany somewhere in the paper.

We added a detailed topographic map (Fig. A1 in the Appendix).

- P8, Section 3.2.  I fail to clearly see a main message appearing from this section; what is the key lesson/result that the authors want to convey?

All existing erosivity maps (mostly unintendedly) employ pronounced smoothing. We explain this now in more detail in the Introduction. Due to our high data availability we were able to and had to replace the unintended and uncontrolled smoothing of existing maps by a statistical protocol because of the large small-scale spatial variability. This chapter is intended (i) to quantify the effects of this protocol, (ii) to justify the protocol and (iii) to highlight the large small-scale spatial variability that even exists for long-term averages. The third point is of particular interest because this large small-scale spatial variability of long-term averages was not known previously due to a lack in suitable data.

- P8L7:  strange combination of present and past tense (smoothed).  Specify that the 10-15km refer to this study.

Changed as recommended

- P8L7-9: the "disappearance" is quite logical from the description of winsorizing.

Yes; we just want to give a measure of the extent of the effect

- P8L11:  "Rain erosivity from 5-min resolution data ..":  it is unclear what erosivity this refers to: annual? Is this for 2011?

Reworded (and the caption of Fig. 3, to which this sentence refers, was improved)

- P8L15-16: probably this is described in methods but a bit unclear why 2011 and 2012 were chosen here.

The only reason is that this data set existed

- P9L32: "extreme" and "violent" sounds rather exaggerated.  A more scientific formulation would be appreciated.

We followed the intensity classification by the UK Meteorological Office, which suggests using 'violent':

"For synoptic purposes, rain showers are classified as 'slight', 'moderate', 'heavy', or 'violent' for rates of accumulation of about 0 to 2 mm h$^{-1}$, 2 to 10 mm h$^{-1}$, 10 to 50 mm h$^{-1}$, or greater than 50 mm h$^{-1}$, respectively."

UK Met Office (2007) Fact Sheet No. 3: Water in the Atmosphere. p. 6. https://web.archive.org/web/20120114162401/http://www.metoffice.gov.uk/media/pdf/4/1/No._03_-_Water_in_the_Atmosphere.pdf

We explicitly refer to this source now.

- P10L17-18: the previous sentences compared radar-based erosivity with meteostation erosivity. Therefore I believe that this sentence makes little sense here, because there may in fact be differences because of the different data used. I more trust the fact that in P10L19 the same stations (and hopefully the same methods) were used. So please do not conclude when the previous statements do not support the conclusion yet.

There are two differences in the Sauerborn approach, first she used meteostation data and second she interpolated via a regression. In L 17-18 we exclude that the regression approach can be the reason for the difference and in L 19 we exclude that the difference in data origin (station/radar) can explain the difference. We hence need both sentences.

- P10L29-30: location is not so clear: could it be indicated in Figure 1?

We added "of the North Sea" to the sentence to make clear where the German Bight is and now we depict "German Bight" and "Baltic Sea" in the map in Fig. 1.

- P11L3-4: perhaps it is me, but I fail to fully grasp what regression was precisely done here (what against what), and with what purpose?

We reworded this sentence; furthermore we explain explicitly in the Introduction now the transfer of point R data to full maps via a regression with precipitation.

- P11L9: "trains" should be "rains"?

This was not a typo. We reworded this sentence ('tracks of thunderstorm movement') to avoid this impression.

- P11L21: see previous points also: I think that this "most pronounced changes" is overstated: the erosivity is still small.

"the most pronounced" is a relative expression; the change may be much smaller than "pronounced"; we left this wording

- P12L28-P13L2: this seems too much repetition of results to me.

We deleted this part

- P13L2: link between sentence ending with "resolutions" and sentence starting with "The pronounced" is not clear; this seems to be another topic.

- P13L3-4 I do not understand this: how can orographic rainfall increase hourly but not the peaks? And with peaks the authors refer to sub-hourly?

We deleted the sentences in L 2-4 because they were speculative anyhow

- P13L16: expected by whom? Rather "than what existing erosivity maps showed"

Reworded

- Captions should be self-explanatory: in Figure 1 I do not understand the "for a range of 128km AND the 2017 configuration". Please revise to make the caption clearer.

We are not sure what is unclear. We added '(utilized radius)' after range indicating that the radar beam does not end but signals of longer travel distances were not used and we replaced 'configuration' by 'tower locations'; note that the following parenthesis explains 'locations of some radar towers have changed over time'

- Figure 2 caption: "Erosivity map" specify in caption if this is annual average erosivity. Also units should be reported in caption.

Wording was changed and units were added.

- Figure 3:  also here it should be clarified if we are looking at annual erosivity, and which years of data and why.  Also why do we see a lag up to 50km?  Would it make sense to make it longer? Why or why not?

Information was added

- Figure 5: caption should specify if daily erosion index (circles and blue) is from radar data.

Information was added

- Figure 6: caption should specify if Sauerborn used the exact same methods

There are likely several differences but these are not documented. E.g., at the time of Sauerborn paper, hyetographs were recorded while nowadays tipping-bucket rain gauges or weighing rain gauges are used. Sauerborn used a manual approach to identify erosive rains, their beginning, their ending and breakpoints of intensity while we used automated calculations. This was done by many and unrelated persons differing in their subjective decisions. We didn't add this because the points are many and their likely effects are unknown. The documentation is rather poor. Insofar, the new map is also a major step forward.

- Table 1:  last two entries (1h winsorized and kriged) are also 17-yr?  Specify.  Also what years are used for annual/biannual?

We added '17-yr'

- Table A2 is probably not needed because Figure 5 is presented. If still kept, it should be organized differently so that Jan-Apr are on one page.

We left Table A2. The figure is easier for the reader (this is why it was inserted in the main body of text) while the Table is required for C factor calculations. We rearranged the table.

- Take care with wording of high/higher/low/lower: usually this refers to altitude, whereas other parameters/values are small or large.

We checked all high/low

---

## Author Comment (AC2) · 10 Dec 2018

The data were noisy so significant data treatment was applied to produce a "typical" erosivity map and an annual cycle of erosivity. The new values show greater erosivity than previously produced maps and the seasonal distribution shows an increase in winter erosivity. The main advantage of the new approach is the use of continuous data over the region.

General comments
The article is very well written, the analyses are clearly described and figures are well chosen. The results will clearly be of use.

However, some of the choices made in data treatment require further justification. My primary concern is about the level of data treatment that has been applied, which is, as the authors state, "extraordinarily large". Because the amount of smoothing applied is indeed more than normal, it should be carefully justified.

All existing erosivity maps (mostly unintendedly) employ pronounced smoothing (presumably much stronger than we did). We explain this now in more detail in the Introduction.
Due to our high data availability we were able to and had to replace the unintended and uncontrolled smoothing of existing maps by a statistical protocol because of the large small-scale spatial variability (which was not known previously due to a lack in suitable data). In contrast to existing maps where the smoothing steps and their effects are largely unknown, we define a statistical protocol, we quantify the effects of this protocol, and we justify the protocol.
Importantly, we also provide the maps without the different smoothing steps in the Appendix and leave it to the decision of the reader, which one to use.

The aim is to produce a map of "typical" erosivity over Germany, but the erosivity distributions in time are skewed and contain outliers (from rare, extreme events) that make finding one representative value per pixel a challenge. A related problem is possible sampling effects, meaning differences between the sampled and true distribution of values (the authors mention this with respect to measurements from gauge networks that may miss entire events). The authors have applied data transformation techniques to find typical values, smooth them in space, and smooth the evolution of the average erosion index over time.

I'd like to comment on each data transformation undertaken, first to produce the per-pixel values:

1. Winsorizing:
For each pixel, the mean erosivity over 17 yearly values is taken using winsorizing. In this case the authors only replace the lowest and the highest value (with the second-lowest and second-highest respectively). How was the choice made to use winsorizing over, say, the sample median? The choice of the method used (e.g. sample median or winsorizing, and the amount of winsorizing used) should be justified – for example through the use of a density plot of erosivity values, in which the skewness will be clearly visible, to show that the final values produced are representative of "typical" values of erosivity.

From a statistical point of view a median or a geometric mean has the advantage of being more robust compared to the arithmetic mean, while the arithmetic mean has the advantage of being unbiased. From a modelling point of view, a median or a geometric mean is inacceptable because modelling individual events would then lead to a different total soils loss than modelling long-term mean soil loss using a robust estimator (due to its bias). Winsorizing was thus an (accepted) method to reduce the effect of extreme outliers (not to reduce the effect of skewness, which is an important property of erosive rains).

The effect of one-step winsorizing, which considers only the most extreme years, was already small. Using a two-step winsorising (replacing the two highest and the two lowest values) would have had an even smaller additional effect.

An acceptable alternative to our approach would have been not to use any winsorizing and kriging but to calculate arithmetic county means (on average 35 km² in size) because county means are likely used in the final application of the map (this is why we also will provide a table with county means). We decided to use the more laborious approach because we expected that would better preserve landscape features smaller than 35 km².

2. Bias-correction:
The authors state that the winsorized mean is biased for long- tailed variables. But, for skewed distributions the winsorized mean should be closer to a "typical value" of the population than the sample mean because of the removal of outlier values. So is it not the case that winsorizing produces a less biased estimate of a central tendancy than the sample mean, and the bias correction suggested by the authors undoes the benefit of the winsorizing by matching back to the (spatial) sample means which are themselves affected by outliers?

The reviewer is right that a winsorized mean should be closer to the expected long-term average. Winsorizing reduces the overweight of an extreme event (upper outlier) at a certain pixel. This reduced overweight should be balanced by reducing the underweight (lower outlier) at a second pixel in order not to cause a bias of the average. For normal distributed variable this works well while for skewed variables this balancing is not fully achieved. This is why we had to correct for the bias in order to arrive at the arithmetic average. This does, however, not cancel the effect of winsoring, because the mismatch in balancing one pixel was not put back to the same pixel but it was distributed among all 455 309 pixel. This is why the bias correction was so small (2.3%) that it could almost be neglected.

3. Ordinary kriging:
Kriging is used to fill gaps not covered by the radar data (due to beam-blocking, for example), and block kriging is used to smooth the output field. Kriging requires at least roughly symmetrically distributed input data (ideally they would be normally distributed) so that mean values are representative. It should therefore be mentioned in the article whether the distribution of "typical" values after the winsorizing procedure is symmetric, and if not whether steps have been taken to correct for this (possible options are a log transformation and/or use of the Cressie variogram estimator). Block kriging is being used in a non-standard way, as a smoother, so that each $1 \times 1$ km$_2$ pixel is estimated as the mean of values across a $10 \times 10$ km$_2$ block. How was the block size of $10 \times 10$ km$_2$ chosen?

Long-term average erosivity does not have the extreme skewness of individual events. The statistical distribution is not normal but may be multimodal because it reflects the contribution

of different landscapes (mountains, plains, coastal regions etc.) to total land. The assumption of normal distribution is even more justified as we use only distances of up to 100 km (i.e., within this distance the data have to be roughly normal distributed). Furthermore, as we do only use kriging as a smoother, the weights of the data would only slightly change with a different semivariogram. A moving window presumably would have done the same job.

The block size was arbitrarily chosen. The entire smoothing steps are rather irrelevant because they do not change the pattern but only provide a slightly smoother map that makes reading easier (please compare the untreated data in Fig. A3 with the final map in in Fig. 2). We provide both maps and leave it entirely to the reader whether he prefers using the unsmoothed Fig. A3 or the smoothed Fig. 2.

After the spatial processing, the annual cycle of erosivity is calculated. Afterwards, smoothing was applied to the daily timeseries of averages. Again commenting on each step:

1. Daily erosion index:
The erosion index is calculated for each pixel and then averaged across space for each day of the year. It was not clear to me whether the pixel values used to make this average were treated in any way (kriged perhaps?) or were raw 1 km$_2$ values (I assume it was the raw values so that they were daily). If the distribution of daily EI values (across space for each day) is heavily skewed, then the mean of their values may not be representative (the median or a winsorized mean, for example, may be better). Was any testing for this done?

Every day was the sum of 17 yr and 455 309 pixel (i.e., n = 7.7 million). To our surprise there was this large scatter between subsequent days (indicating that 17 yr are still short and that pixel cannot replace time). We did not remove any data because a high average value with n = 7.7 million can only happen, if very many pixels at that day had a high value (i.e., erosive events occurred in many regions at the same day; this is the opposite of an outlier). The lowest value is zero and it can also not be removed because this would be the majority of all days (days without an erosive event).

The main question is whether the daily erosion index varies among landscapes. In order to analyse this, we added in the revision the daily erosion index for the SE, SW, NE and NW quadrant of Germany. These four quadrants differ in climatological properties (e.g., continentality; annual rainfall and altitude) but the resulting pattern was identical for all quadrants (correlation between them yielded r² > 0.998) indicating a stability of the seasonal pattern. This close correlation also indicates that the influence of outliers is small.

2. 13-day centred median, 3-day skip mean, and 25-day centred hanning mean:
This choice of smoothing routines needs to be better justified. Why was this combination of window sizes (13, 3, and 25 day) and operators chosen, and how was it judged whether the smoothed values represented the true signal? As a suggestion that may provide more information to the reader: the authors could consider displaying maps not only of winsorized mean of annual values, but also per- pixel median, 10th percentile, and 90th percentile, to show not only "typical" values but maps of extreme erosivity values as well, and to show the spread of values for each pixel.

We added a paragraph on smoothing in the Introduction.

In short, the statistical recommendation is to use smoothing until the information can be seen that is intended to be shown. This recommendation may unsatisfactory for the inexperienced

and appears arbitrary. However, there cannot be a statistical criterion because the degree of required smoothing depends on the intended application, which is outside statistics.

We could have used a cumulative density function (cdf) instead of a probability density function (pdf) to describe seasonality. This would have required no smoothing at all but with the cdf it would have been more difficult for the user to assess the effect of different crop management options. The cdf of the unsmoothed data correlates with the cdf of the smoothed data with $r^2$=0.9998 (n=365), proofing that smoothing has not caused any relevant change in seasonality. We added to the manuscript:

"Despite the strong smoothing that was necessary for the probability density function, the smoothing did not change the cumulative density function (which is used for calculating C factors). The cumulative density functions of the original data and of the smoothed data correlated with $r^2$=0.9998 (n=365)."

The window sizes are typically used values. A median with n=2 would produce an arithmetic mean. The advantage of the median filter compared to the arithmetic mean increases with increasing window size. At a window size of n=365 identical values for all days would result and the seasonality would be completely destroyed. A window size of n=13, which is also about the time window of individual crop management practices, ascertains that the seasonality is maintained. A skip mean is always applied with n=3. For the hanning mean a similar reasoning can be put forward as for the median filter. Due to the decreasing weights, the window size can be somewhat larger than for the median filter and window size has less influence (even for n=365 some seasonality would be retained). Also the order of the filters is quasi standard. A median filter only makes sense if it is applied in first place. After some averaging has occurred, the median filter loses its advantage compared to the arithmetic mean and both would produce very similar values. The skip mean cannot be the last filter because it distributes a high (or low) value completely to its neighbours and a high (or low) value is replaced by the low (or high) values of the neighbours. The skip mean thus inverts the pattern within the window size of n=3. Hence, it has to be followed by some averaging, which puts the high (or low) value back in place because during averaging some information is inherited from the neighbours on both sides. The hanning mean is especially versatile in this case because the weights (or the window size) allow adjusting to some degree how much of the high (or low) values is put back in place and how much is left with the neighbours.

Again, the statistical treatment is not critical because we also display the original data. The reader can decide whether he likes our smoothing or not. He may draw his own line where he thinks that the line should be (this is how it was done by Wischmeier and Smith, 1978, and still by Rogler and Schwertmann, 1981).

Specific comments
1. Page 2, line 24–25: "Unstable and unreliable transfer functions result that differ pronouncedly" – I do not understand the sentence, could you please rephrase?

Rephrased

2. Page 2, line 30: Please include a general reference for the radar measurement principle. One such reference could be the book by Bringi and Chandrasekar (2001), Polarimetric Doppler Weather Radar, Cambridge Uni. Press.

Thank you for the suggestion. To put this into the European context, we additionally cite:

Meischner, P., Collier, C., Illingworth, A., Joss, J., Randeu W.: Advanced weather radar systems in Europe: The COST 75 action, Bull. Amer. Meteorol. Soc., 78, 1411-1430, 1997.

3. Page 3, line 12: I see your point that the use of continuous radar data avoids missing large and rare events that could be missed completely by gauge networks. But you do a lot of processing to the radar data, including winsorizing and smoothing, which reduces the influence of rare extreme events on the summary statistics. There are two separate problems here: sampling (gauges may miss an event) and then what value to use as a "typical" value for a skewed distribution. It is important that justifications for the data treatment show that the chosen "typical" measure is appropriate.

We fully agree. This is why we examine in detail the effects of the different statistical steps during smoothing. Now we also provide in the Introduction an overview over the previously used, unintended and uncontrolled steps of data manipulation that were necessary to arrive from gauge data at erosivity maps. The origin of the problem, however, rests in the characteristics of erosive events and can thus not be avoided. The use of contiguous radar data allows replacing several poorly controlled steps by clearly defined statistical methods, which is a major step forward but certainly better methods will be found in future the better we understand the origin of variation, the more data are gathered and the better the computing methods will become.

4. Page 3, line 31: Which Z-R relationship is used?

We provide a reference now

5. Page 4, line 6: Is the figure of 1 gauge per 80 $km_2$ an average value?

Yes; we added "equivalent to" to make clear that this density is calculated from the number of 4000 stations per Germany

6. Page 4, line 20: For clarity, it would be helpful to include the units of $I_{max}30$ and $E_{kin}$ when the variables are introduced here; this is especially important because $E_{kin}$ [kJ $m_{-2}$] and $E_{kin,i}$ [kJ $m_{-2}$ $mm_{-1}$] have different units.

We included the units when the variables were introduced

7. Page 4, line 24: You should reference Fischer et al 2018 (from your references list) here since your definitions, units, and descriptions are very similar to those used in your previous paper.

We cite Rogler and Schwertmann (1981) now because they were the first who transferred the Wischmeier equations in US units to SI units.

8. Page 4, line 29: "the $R_e$ sum" – do you mean "the sum of $R_e$"?

Yes; we rephrased the sentence

9. Page 5, lines 14–15: For a given pixel, if too many years were excluded then the sampling may become less representative. How often were pixels affected by this exclusion of years, and were there pixels for which many years were excluded?

The requested information had already been given ("If the effective number of excluded years was larger than one, the respective pixel was excluded. This was the case for 0.6% of all pixels.") but now we moved it to a different place where it may be expected more by the reader.

This loss of years mainly occurred in marginal regions in the very North or very South that had only been captured by radar data before or after the displacement of radar sites (e.g. in the far North due to the shift of the radar Hamburg to Boostedt in 2014).

10. Page 5, lines 16–22: "replaced by the maximum 1-h rain depth" - should this read rain intensity?

Usually rainfall is reported for a certain period of time (per day/month/year) and hence this should always be called intensity instead of depth. However, depth is commonly used if longer periods of time (days, years) are considered during which short-term intensity varies.

We replaced "depth" by "intensity".

11. Page 5, lines 16–22: As I understand it, the scaling factors are being used to adjust the method of calculating erosivity to put a "virtual rain gauge" in each radar pixel, to account for the fact that radar measurements are areal and integrated over time and therefore smooth out rainfall intensity peaks. Since rain intensity depends on temporal resolution, and you require 30 minute maximum rain rates which would be smoothed by the use of 1 hour radar data, I see why a temporal scaling factor could be used. But spatially, the areal measurements at 1 $km_2$ resolution can be assumed to be representative of each 1 $km_2$ pixel, and since you are producing erosivity values at the same resolution, I don't understand why a spatial scaling factor (or indeed the method scaling factor) is required. Please could you explain more here why the scaling factors are used and how they are applied (e.g. it is not clear which threshold is lowered to 5.8 mm $h_{-1}$).

We agree that this is difficult to understand. We had to dedicate an entire publication to this tricky question (Fischer et al., 2018). An additional explanation with one or two sentences in this manuscript would open more questions that it would answer. We prefer to leave it like this.

In the discussion we now explain (and cite a reference) that rain erosivity on average drops to one fourth at only 2 km distance from the centre of a convective rain cell. This pronounced patchiness requires a spatial scaling factor.

We repeated "Imax30" in this sentence to make clear, which threshold had to be adjusted.

12. Page 6, line 14: The use of some independent data to test the spatial representativity of the smoothed data is a good idea, but is this test data independent? It is also based on radar data. Has the test region data been compared to gauges or other ground truth data to ensure it is accurate?

In this case we only wanted to examine the effects of our data treatment but not the comparison between radar-derived and gauge-derived erosivities. We have done this extensively in Fischer et al. (2018), which we cite. In this publication we compared 19 944 events observed at 115 stations with radar-derived erosivities for the same location.

13. Page 7, lines 1–2: "The cumulative distribution curve for the test region calculated from 5-min data will then be a fair estimate of the return periods anywhere in the research area" – I do not think this is proven. Even if the test region and the whole area agree at 1 hour resolution, extreme intensities are smoothed out at this lower resolution, so it does not necessarily follow that the 5 minute cumulative distributions are the same across all regions.

We weakened the statement:

"The cumulative distribution curve for the test region was also calculated using the 5-min rain data. Given that the cumulative distribution curves of the entire study area and the test region agree for the relative erosivities calculated from 1-h data, we expect that the relative erosivities calculated from 5-min rain data of the test region can serve as a first estimate for the entire study region."

14. Page 7, lines 19–20: Please include a reference for these statements about radar accuracy (they are correct but require a citation).

We added a reference

15. Section 3.1: I suggest that to back up your observation that the regional pattern in erosivity is dominated by orography, you should include a topographic map showing ground elevation for comparison with the map in Figure 2.

We added a detailed topographic map in the Appendix

16. Page 8, line 13: "Using the normal distribution" – but are the erosivity values normally distributed?

This is clearly not the case although long-term average erosivity does not have the extreme skewness of individual events or the skewness of individual years. The statistical distribution is not normal but may be multimodal because it reflects the contribution of different landscapes (mountains, plains, coastal regions etc.) to total land. The assumption of a normal distribution is only used here for illustration purposes to "translate" the semivariance, which may be difficult to grasp by some readers, to the ordinary R factor space. This assumption has no further relevance for our analysis. We deleted this phrase.

17. Page 8, line 32: Which variogram was the kriging conditioned by? I would expect kriging to maintain the spatial stucture even in a block kriging case, so it is odd that the kriging changes the variogram.

The semivariogram used for kriging is indicated as "1 h, 17 yr" in Fig. 3. We added a numbering to the semivariograms in Fig. 3 so that we can clearly refer to each of the semivariograms in the text.

The semivariogram after kriging always differs from the semivariogram before kriging because at least the nugget effect is removed by kriging. Block kriging will additionally remove any pattern that is smaller than the block size. It can then not appear anymore in the semivariogram after kriging. This is a well-known phenomenon in geostatistics and usually addressed as change of support or as regularization or as modifiable areal unit problem. Such a change of support may even turn a spherical model into a Gaussian model. See for instance:

Clark I.: Regularization of a semivariogram, Computers & Geosciences, 3, 341-346, 1977.

Gotway, C.A., Young, L.J.: Combining incompatible spatial data, J. Amer. Statistical Assoc., 97, 632-648, 2002.

Emery, X.: On some consistency conditions for geostatistical change-of-support models, Mathematical Geology, 39, 205-223, 2007.

18. Section 3.3: I suggest adding more lines to your Figure 4 to show all the lines you mention in the text.

We do not understand this comment. In the text we mention return periods of 2 yr, 30 yr and 100 yr. In Fig. 4 we show lines indicating return periods of 2 yr, 5 yr, 10 yr, 30 yr and 100 yr.

19. Page 9, line 21: I'm surprised that you would expect less than the mean erosivity for an event with a return period of 2 years. Any comment there?

This is due to the fact that total erosivity is determined by extreme events. To illustrate this: the largest event recorded by Fischer et al (2018) during only two months was 1270 N h$^{-1}$. This means that if at that location no other erosive rain would fall during the next 10 yr, the average annual rain erosivity would still be the same as that of all other locations.

20. Page 9, line 27: "d" is presumably for days but should be spelled out.

This would not comply with the recommendations of NIST and other institutions of standards, which require that unit symbols and unit names must not be mixed.

Thompson, A., Taylor, B.N., Guide for the Use of the International System of Units (SI), NIST Special Publication 811, 2008

21. Page 9, line 32: To see exactly what is going on here, did you compare the distributions of erosivity values for each of these example days? I suspect that the median values would be more stable.

The value was calculated as the accumulated erosivity of an individual day over 17 yr and all pixels divided by the accumulated erosivity over all days and all pixels during 17 yr. Hence there was only one value for each day. Calculating the erosivity index individually for each year and each pixel would yield a (very stable) median of zero for all days (because there are only about 20 events per year at a location).

What was surprising to us was the fact that despite this huge sample size for every day (almost 8 million pixel days), the values still differed so much for consecutive days. This is because averaging over years cannot be replaced by averaging over locations. During some

days many locations (pixels) received erosive rainfall while on the next day the weather system may have changed and no pixel receive erosive rain. It is clear that with 20 events per year and 17 years (=340 events per pixel) some days must exist in each pixel that cannot have received an event even if the events would be evenly distributed.

22. Page 10, line 20: I think Fig. 5 should be Fig. 6.

We corrected the typo

23. Page 11, line 18: No definition of C-factor calculations is given; please add one.

We added a reference here. A short description was already given in the Introduction

24. Page 12, line 22: Please define (R)USLE.

RUSLE is explained in the Introduction now

25. Figures 2, A1, and A2: Units for the plotted variable should be stated either in the key or caption.

Information was added

Technical corrections
• Page 5, line 19: The word "occurred" can be removed.

Sentence was rephrased

• Page 7, line 12: By "in the smooth" do you mean "in the smoothing operation"?

This wording had been taken from the cited statistical reference. Now we reworded the sentence and used common language.

• Figure 2: In the caption the average sizes of the local authority and community areas should be areas (km$_2$).

Changed as requested

• Page 8, line 8: "very extreme" is redundant when "extreme" will do.

Changed as requested

• Page 9, line 23: "extremer" should be replaced by "more extreme".

Changed as requested

---

## Author Response (AR2)

Dear Editor:

Below, you will find all comments that we received and in blue we inserted how we considered the advice in our manuscript. Additionally we have carefully checked the manuscript for typos and other formal weaknesses.

Please note that we have changed the order of authors and that we added a fourth author.

Karl Auerswald (on behalf of the authors)

**Editor Decision: Publish subject to minor revisions (review by editor)** (01 Feb 2019) by Remko Uijlenhoet
Comments to the Author:
Dear authors,

The two reviewers have a few minor editorial suggestions, which I urge you to take into account when preparing the final version of your manuscript.

In addition, one of the reviewers noticed that the wording in Section 2.2 is similar to a section in a previous paper by you (hess-22-6505-2018). The section introduces generic concepts and is obviously hard to word differently every time, but one sentence is almost word-for-word the same as in the previous paper ("that rain events stop and start on average in the middle of the first and the last nonzero rain interval"). Please try to rephrase this sentence a bit.

We rephrased this sentence significantly

Report #1
Referee #1: Anton Vrieling, a.vrieling@utwente.nl

**Suggestions for revision or reasons for rejection (will be published if the paper is accepted for final publication)**
The authors have thoroughly responded to my concerns and updated the manuscript accordingly. I am happy with their responses and with their important and interesting contribution. In my opinion, the manuscript can be published pending a few minor corrections/edits that may be implemented at the authors' discretion.

- P1L24: as the statement on winter wheat is not directly related to this study, I suggest removing this sentence (or alternatively write as "For example, …", although my preference is removal).

We rephrased the sentence

- P2L28: "and" between "zero" and "erosivity" should be removed.

We deleted the erroneous "and"

- P3L11: I suggest not starting the paragraph with "Also" but rather rephrase to: "Existing R maps have also undergone a number of …"

Rephrased as suggested

- P3L11: the subsentence "which were rather uncontrolled and of unknown degree" remains
  vague. Probably the authors want to convey something like: "even if this is not
  explicitly stated in the corresponding reports".

We revised the sentence as suggested

- P3L20 "as geostatistical tools existed not yet". I think that this will not hold in court! In
  the early 80s we certainly had geostatistics with uses among others in mining.
  Perhaps rather: "as geostatistical tools were not yet commonplace in various
  application fields"

We rephrased the sentence although it was correct. It referred to the last normal-period
rainfall map (1931 to 1960), which was well before Matheron's founding treatise in 1970
„La theorie des variables regionalisees et ses applications"

- P3L21-22: I would suggest to combine this sentence with the previous paragraph, and
  delete the rest (P3L23-P3L27) of that paragraph (or maybe integrate in methods). In
  my view it is too much 'fast-forwarding' to the methods section, whereas the main
  objective of the paper has not been expressed at that point.

We combined both paragraphs and deleted this part of the paragraph as suggested.

- P5L28: I note that this is just a single station in Belgium and the "not deviate" is perhaps
  in fact a small deviation (not fully clear to which results by van Dijk this relates?). I
  was thinking to replace "in Belgium" here with "for the nearest European station
  used in Belgium" (or perhaps to include also the Italian station, also within
  reasonable vicinity of southern Germany?)

We wouldn't expect a similarity with an Italian station (even though it might exist) because
of the Mediterranean climate that cannot be found even in southern Germany. We didn't
change the sentence because in the previous sentence we had already pointed out that
Wilken et al. (a study from Germany) also found no reason to reject the Wischmeier
equation.

- P15L23-24: I trust that the full links will be added.

We completed the data availability section. The final map and maps of all 17 individual
years can be downloaded from two locations at the Climate Data Center of the German
Weather Service.

- P15L25: "until" should probably read "by"? Can anything be said regarding delivering of
  these erosivity maps for years after 2019?

We added: „Annual maps of future years will routinely be produced and published within
the framework of the annual RADKLIM update after the precipitation data have undergone
all steps of quality control and refinement."

Anton Vrieling

Report #2
Anonymous Referee #2

**Recommendation to the Editor**

**Suggestions for revision or reasons for rejection (will be published if the paper is accepted for final publication)**

I thank the authors for taking the reviewer comments into account and for the improvements they have made to the manuscript. With only a couple of technical changes (listed below) the manuscript is now ready for publication. It will make a valuable contribution to the literature.

Technical corrections:

* Page 14, line 2, Figure 5A should read Figure A5.

We corrected the typo

* Page 15, line 1, "Another implication of this large variability is that 20 yr will still not be sufficient" -- is this conclusion drawn from the 17 yr data set and the processing it required? Please rephrase to make the basis for the claim more clear.

We explained this conclusion better by referring to an example

* The data availability section is incomplete.

We completed the data availability section (details see above)

[revised manuscript text omitted]

---

## Author Response (AR3)

Editor:

**However, I noticed that you changed the order of the authors of the paper (the 3rd author on the previous version of the paper became the lead author) and you added a 4th author. Before I can proceed with accepting your paper, I need to have a clear and convincing motivation why these changes in the authorship were proposed. This also begs the question why you did not use your final list of authors in the original version of the paper.**

Reply:

Dear Editor:

The reasons leading to the original list of were mainly to promote the PhD student (F. Fischer) who had done large parts of the analysis. The principal investigator of the project (R. Brandhuber) was only mentioned in the Acknowledgements. During writing and even more during revision it turned out that most of the work load rested on the corresponding author (listed as senior author). We hence decided to add the principal investigator as senior author and to move the former senior author in the front position.

We usually quantify the contributions of authors according to the following seven categories: concept / funds / data / analysis / background and interpretation / writing / revision. Averaged over these categories (with data analysis being weighted more in this case) the contribution of authors was as follows: KA 48% (equally contributing to all categories), FF 22% (mainly contributing to data analysis), TW 24% (mainly providing the data), RB 4% (mainly providing the funding). The remaining 3% were contributed by four other persons, three of them being mentioned in the Acknowledgements.

Karl Auerswald

---

## Author Response (AR4)

Dear Editor:

We resubmit our manuscript with an unchanged authors' list. As outlined in several emails, we wish to keep this list. The question is sufficiently answered, why we find this list appropriately reflecting the contributions and responsibilities. The question, why we originally had chosen another order of authors that did not reflect these contributions is historic and thus irrelevant for the present manuscript.

Karl Auerswald, on behalf of the authors